**Floodwater Impact on Galveston Bay Phytoplankton Taxonomy, Pigment Composition and Photo-**
**Physiological State following Hurricane Harvey from Field and Ocean Color (Sentinel-3A OLCI)**
**Observations**

**Bingqing Liu[1], Eurico D'Sa[1*] and Ishan Joshi[1]**
[1]Department of Oceanography and Coastal Sciences, Louisiana State University, Baton Rouge, LA 70803,
USA
* Corresponding author: ejdsa@lsu.edu
**Abstract**
Phytoplankton taxonomy, pigment composition and photo-physiological state were studied in Galveston
Bay (GB), Texas (USA) following the extreme flooding associated with Hurricane Harvey (August 25–29,
2017) using field and satellite ocean color observations. Percentage of chlorophyll a (Chl a) in different
phytoplankton groups were determined from a semi-analytical IOP (inherent optical property) inversion
algorithm. The IOP inversion algorithm revealed the dominance of freshwater species (diatom,
cyanobacteria and green algae) in the bay following the hurricane passage (September 29, 2017) under
low salinity conditions associated with the discharge of floodwaters into GB; 2 months after the hurricane
(October 29–30, 2017), under more seasonal salinity conditions, the phytoplankton community
transitioned to an increase in small sized groups such as haptophyte and prochlorophyte. Sentinel-3A
OLCI-derived Chl a obtained using a red/NIR band ratio algorithm for the turbid estuarine waters was
highly correlated ($R^2 > 0.90$) to HPLC-derived Chl a concentrations. Long-term observations of OLCI-
derived Chl a (August, 2016-December, 2017) in GB revealed that hurricane-induced Chl a declined to
background mean state in late October, 2017. A Non-Negative Least Square (NNLS) inversion model was
then applied to OLCI-derived Chl a maps of GB to investigate spatiotemporal variations of phytoplankton
diagnostic pigments pre- and post-hurricane; results appeared consistent with extracted phytoplankton
taxonomic composition derived from the IOP inversion algorithm and microplankton pictures obtained
from an Imaging FlowCytobot (IFCB). OLCI-derived diagnostic pigment distributions also exhibited
good agreement with HPLC measurements during both surveys, with $R^2$ ranging from 0.40 for
diatoxanthin to 0.96 for Chl a. Environmental factors (e.g., floodwaters) combined with phytoplankton
taxonomy also strongly modulated phytoplankton physiology in the bay as indicated by measurements of
photosynthetic parameters with a Fluorescence Induction and Relaxation (FIRe) system. Phytoplankton in
well-mixed waters (mid-bay area) exhibited maximum PSII photochemical efficiency ($F_V/F_M$) and low
effective absorption cross section ($\sigma_{PSII}$), while the areas adjacent to the shelf (likely nutrient-limited)
showed low $F_V/F_M$ and elevated $\sigma_{PSII}$ values. Overall, the approach using field and ocean color data
combined with inversion models allowed, for the first time, an assessment of phytoplankton response to a
large hurricane-related floodwater perturbation in a turbid estuarine environment based on its taxonomy,
pigment composition and physiological state.
**Key words**: Galveston Bay, phytoplankton taxonomy, pigment composition, physiology, ocean color,
chlorophyll a, Sentinel-3A OLCI

## 1. Introduction

Phytoplankton, which form the basis of the aquatic food web, are crucial to marine ecosystems and play a strong role in marine biogeochemical cycling and climate change. Phytoplankton contributes approximately half of the total primary production on Earth, fixing ~50 GT of carbon into organic matter per year through photosynthesis; however, various phytoplankton taxa affect differently the carbon fixation and export (Sathyendranath et al., 2014). Chlorophyll a (Chl a), an essential phytoplankton photosynthetic pigment, has been considered a reliable indicator of phytoplankton biomass and primary productivity in aquatic systems (Bracher et al., 2015). Phytoplankton also contain several accessory pigments such as chlorophyll-b (Chl b), chlorophyll-c (Chl c), photosynthetic carotenoids (PSC) and photo-protective carotenoids (PPC) that are either involved in light harvesting, or in protecting Chl a and other sensitive pigments from photo-damage (Fishwick et al., 2006). Some of PSCs and PPCs are taxa-specific and have been considered as bio-marker pigments: e.g., fucoxanthin (PSC) for diatoms, peridinin (PPC) for certain dinoflagellates, alloxanthin (PPC) for cryptophytes, zeaxanthin (PPC) for prokaryotes (e.g. cyanobacteria), and the degradation products of Chl a, namely, divinyl Chl a and divinyl-Chl b for prochlorophyte (Jeffrey and Vest, 1997). High-Performance Liquid Chromatography (HPLC) which can efficiently detect and quantify several chemo-taxonomically significant chlorophylls and carotenoids, when coupled with these taxa-specific pigment ratios, allow phytoplankton taxonomic composition to be quantified based on a pigment concentration diagnostic procedures such as CHEMTAX (Mackey et al., 1996). Furthermore, phytoplankton pigments with distinct absorption characteristics strongly influence the light absorption by phytoplankton (Bidigare et al., 1990; Ciotti et al., 2002; Bricaud et al., 2004). As such, phytoplankton absorption spectra have been used to infer underlying pigments including phytoplankton taxonomy by Gaussian-decomposition (Hoepffner and Sathyendranath, 1991; Lohrenz et al., 2003; Ficek et al., 2004; Chase et al., 2013; Moisan et al., 2013; Wang et al., 2016; Moisan et al., 2017). More importantly, phytoplankton optical properties (absorption and backscattering) bearing the imprints of different pigments and cell-size, are important contributors to reflectance in a waterbody (Gordon et al., 1988). Morel and Prieur, (1977) first reported the feasibility of calculating the phytoplankton absorption coefficients and other inherent optical properties (IOPs) from measured subsurface irradiance reflectance based on the simplified radiative transfer equation. Improvements in semi-analytical inversion algorithms to derive IOPs from in-situ and remotely sensed reflectance spectra have been reported (Roesler and Perry, 1995; Hoge and Lyon, 1996; Lee et al., 1996; Garver and Siegel, 1997; Carder et al., 1999; Maritorena et al., 2002; Roesler and Boss, 2003; Chase et al., 2017). Roesler et al. (2003) further modified an earlier IOP inversion algorithm used in Roesler and Perry, (1995) by adding a set of 5 species-specific phytoplankton absorption spectra, and derived phytoplankton taxonomic composition from the field measured remote sensing reflectance.

Phytoplankton pigment composition varies not only between taxonomic groups but also with photo-physiological state of cells and environmental stress (e.g., light, nutrients, temperature, salinity, turbulence and stratification) (Suggett et al., 2009). The photosynthetic pigment field is an important factor influencing the magnitude of fluorescence emitted by phytoplankton, with active fluorometry commonly used to obtain estimates of phytoplankton biomass (D'Sa et al., 1997). Advanced active fluorometry termed as fast repetition rate (FRR; Kolber et al., 1998) and analogous techniques such as the fluorescence induction and relaxation (FIRe; Suggett et al., 2008) allows for the simultaneous measurements of the maximum PSII photochemical efficiency ($F_v/F_m$; where $F_m$ and $F_o$ is maximum and minimum fluorescence yield and $F_v$ is variable fluorescence obtained by subtracting $F_o$ from $F_m$) and the effective absorption cross section ($\sigma_{PSII}$) of a phytoplankton population; these have been used as diagnostic indicators for the rapid assessment of phytoplankton health and photo-physiological state linked to environmental stressors. Considerable effort has been invested to achieve a deeper understanding of the impacts of environmental factors and phytoplankton taxonomy on photosynthetic performance of natural communities from field and laboratory fluorescence measurements (Kolber et al., 1988; Geider et al., 1993; Schlüter et al., 1997; Behrenfeld and Kolber, 1999; D'Sa and Lohrenz, 1999;

Holmboe et al., 1999; Moore et al., 2003). Furthermore, knowledge of photo-physiological responses of
phytoplankton in combination with information on phytoplankton taxonomic composition could provide
additional insights on regional environmental conditions.
Synoptic mapping of aquatic ecosystems using satellite remote sensing has revolutionized our
understanding of phytoplankton dynamics at various spatial and temporal scales in response to
environmental variabilities and climate change. It has also provided greater understanding of biological
response to large events such as hurricanes in oceanic and coastal waters (Babin et al. 2004; Acker et al.
2009; D'Sa 2014; Farfan et al. 2014; Hu and Feng, 2016). Although the primary focus of ocean color
sensors has been to determine the Chl a concentration and related estimates of phytoplankton primary
production (Mitchell, 1994; Behrenfeld and Falkowski, 1997), more recently, several approaches have
been developed based on phytoplankton optical signatures to derive spatial distributions of phytoplankton
functional types (PFTs) (Alvain et al., 2005; Nair et al., 2008; Hirata et al., 2011), phytoplankton size
classification (Ciotti et al., 2002; Hirata et al., 2008; Brewin et al., 2010; Devred et al., 2011), and
phytoplankton accessory pigments (Pan et al., 2010; Pan et al., 2011; Moisan et al., 2013; Moisan et al.,
2017; Sun et al., 2017). The basis of these satellite-based remote sensing algorithms have relied on
distinct spectral contributions from phytoplankton community composition (e.g., taxonomy, size structure)
to remote sensing reflectance ($R_{rs}$, sr$^{-1}$); however, these studies have all been confined to open ocean and
shelf waters. In contrast, satellite studies of phytoplankton pigments have been limited in the optically
complex estuarine waters where the influence from wetlands, rivers, and coastal ocean make
phytoplankton communities highly variable and complex.
In this study, field bio-optical measurements and ocean color remote sensing data (Sentinel-3A OLCI)
acquired in Galveston Bay, a shallow estuary along the Gulf coast (Texas, USA; Fig. 1), are used to
investigate the spatial distribution of phytoplankton pigments, taxonomic composition, and their photo-
physiological state following the extreme flooding of the Houston Metropolitan and surrounding areas
due to Hurricane Harvey and the consequent biological impact of the floodwater discharge into the bay.
The paper is organized as follows: section 2 describes the field data acquisition and laboratory processing,
section 3 presents the algorithms and methods used to distinguish phytoplankton groups, retrieve spatial
distribution of pigments, and calibrate phytoplankton physiological parameters. Results and discussions
(section 4 and 5), and summary (section 6) addresses the main contributions and findings of this paper.
**2.   Data and Methods**
**2.1 Study area**
Galveston Bay (GB), a shallow water estuary (~2.1 m average depth), encompasses two major sub-
estuaries: San Jacinto Estuary (also divided as Upper GB and Lower GB), and Trinity Estuary (Trinity
Bay) (Fig. 1). It is located adjacent to the heavily urbanized and industrialized metropolitan areas of
Houston, Texas (Dorado et al., 2015). A deep (~14 m) narrow Houston Ship Channel connects the bay to
the northern Gulf of Mexico (nGoM) through a narrow entrance, the Bolivar Roads Pass. Tidal exchange
between GB and the nGoM occurs through the entrance channel with diurnal tides ranging from ~0.15 to
~0.5 m. The major freshwater sources to GB are the Trinity River (55%), the San Jacinto River (16%),
and Buffalo Bayou (12%) (Guthrie et al., 2012). The San Jacinto River was frequently observed to
transport greater amounts of dissolved nutrients into GB than the Trinity River (Quigg, 2011); however,
the negative relationship between nitrate concentrations and salinity observed in the mid-bay area
(adjacent to Smith Point) (Santschi, 1995), indicated Trinity River to be a major source of nitrate in GB.
The catastrophic flooding of Houston and surrounding areas associated with Hurricane Harvey resulted in
strong freshwater inflows into GB from the San Jacinto River (>3300 m$^3$s$^{-1}$; USGS 08067650) on August
29, 2017 and the Trinity River (>2500 m$^3$s$^{-1}$; USGS 08066500 site at Romayor, Texas) on August 30,
2017, respectively. Although the discharge from the two rivers in the upstream returned to normal
conditions (~50–120 m$^3$s$^{-1}$) in about 2 weeks after the Hurricane passage, salinity remained low for over a
month following the hurricane passage (D'Sa et al., 2018).

## 2.2 Sampling and Data Collection

Surface water samples were collected at a total of 34 stations during two surveys on September 29 and
October 29–30, 2017 (Fig. 1). Samples at stations 1 to 14 (red asterisk on top of green circle; Fig. 1)
along the Trinity River transect were collected repeatedly on September 29 and October 29, 2017,
respectively. Additional 9 sampling sites (blue squares; Fig. 1) around the upper bay and in the East Bay
were sampled on October 30, 2017. The surface water samples were stored in coolers and filtered on the
same day. The filter pads were immediately frozen and stored in liquid nitrogen for laboratory absorption
spectroscopic and HPLC measurements of the samples. An optical package equipped with a conductivity-
temperature-depth recorder (Sea-Bird SBE) and a Fluorescence Induction and Relaxation System (FIRe;
Satlantic Inc) was used to obtain profiles of salinity, temperature, pressure, and phytoplankton
physiological parameters ($F_V/F_M$ and $\sigma_{PSII}$). Measurements of backscattering were also made at each
station using the WETLabs VSF-3 (470, 530, 670 nm) backscattering sensor (D'Sa et al. 2006). Included
in the optical package was also a hyperspectral downwelling spectral irradiance meter (HyperOCR,
Satlantic). The irradiance data from HyperOCR were processed using Prosoft 7.7.14 and the
photosynthetically Active Radiation (PAR) were estimated from the irradiance measurements.   The
above-water reflectance measurements were collected using a GER 1500 512iHR spectroradiometer in
the 350-1050 nm spectral range.  At each station, sky radiance, plate radiance, and water radiance were
recorded (each repeated three times) and processed to obtain above-water remote sensing reflectance
(Joshi et al., 2017). A total of 43 Sentinel-3A OLCI full resolution mode, cloud free level-1 images were
obtained over GB between August 01, 2016-December 01, 2017 from the European Organization for
Meteorological Satellites (EUMESAT) website and pre-processed using Sentinel-3 Toolbox Kit Module
(S3TBX) version 5.0.1 in Sentinel Application Platform (SNAP). These Sentinel-3A OLCI data were
further atmospherically corrected to obtain remote sensing reflectance ($R_{rs\ OLCI}$, sr$^{-1}$) using Case-2
Regional Coast Color (C2RCC) module version 0.15 (Doerffer and Schiller, 2007). River discharge
information during August, 2016-December, 2017 was downloaded from the USGS Water Data (USGS)
for Trinity River at Romayor, Texas (USGS 08066500) and the west flank of the San Jacinto River
(USGS 08067650). Individual pictures of microplankton (10 to 150 µm) recorded by an Imaging
FlowCytobot (IFCB) located at the entrance to Galveston Bay were downloaded (http://dq-cytobot-
pc.tamug.edu/tamugifcb) for pigment validation.

## 2.3 Absorption Spectroscopy

Surface water samples were filtered through 0.2-µm nuclepore membrane filters and the colored dissolved
organic matter (CDOM) absorbance ($A_{CDOM}$) were obtained using a 1-cm path length quartz cuvette on a
Perkin Elmer Lambda-850 UV/VIS spectrophotometer equipped with an integrating sphere. The
Quantitative Filter Technique (QFT) with 0.7-µm GF/F filters were used to measure absorbance of
particles ($A_{total}$) and non-algal particles ($A_{NAP}$) inside an integrating sphere at 1 nm intervals from 300 to
800 nm. The absorption coefficients of CDOM ($a_{CDOM}$), NAP ($a_{NAP}$), particles ($a_{total}$) and phytoplankton
($a_{PHY}$) were calculated using the following equations:

$$a_{CDOM}(\lambda) = 2.303 \times \frac{A_{CDOM}(\lambda)}{L} \qquad \dots\dots (1)$$

where L is the path length in meters. The $a_{CDOM}$ were corrected for scattering, temperature, and baseline
drift by subtracting an average value of absorption between 700-750 nm for each spectrum (Joshi and
D'Sa, 2015).
$$a_{total}(\lambda) = 2.303 \times \frac{A_{total}(\lambda)}{V_{filtered}/S_{filter}} \qquad \ldots\ldots (2)$$

$$a_{NAP}(\lambda) = 2.303 \times \frac{A_{NAP}(\lambda)}{V_{filtered}/S_{filter}} \qquad \ldots\ldots (3)$$

$$a_{PHY}(\lambda) = a_{total} - a_{NAP} \qquad \ldots\ldots (4)$$

where $V_{filtered}$ is the filtered volume of sample, $S_{filter}$ is the area of filter pads and the path length correction
for filter pad was applied according to (Stramski et al., 2015).
**2.4 Pigment Absorption Spectra**
The water samples were filtered with 0.7-µm GF/F filter. The filter pads were stored in liquid nitrogen
until transferred into 30 ml vials containing 10 ml cold 96% ethanol (Ritchie, 2006). The vials were spun
evenly to ensure full exposure of the filter pad to the ethanol and then kept in the refrigerator (in the dark)
overnight. The pigment solutions at room-temperature were poured off from vials into 1 cm cuvette and
measured on a Perkin Elmer Lambda-850 UV/VIS spectrophotometer to obtain pigment absorbance
spectra ($A_{pig}$), while, 90% ethanol was used as a blank (Thrane et al., 2015). The total absorption
coefficients of pigments $a_{pig}(\lambda)$ were calculated as follow:
$$a_{pig}(\lambda) = 2.303 \times \frac{A_{pig}(\lambda)}{L} \times (\frac{V_{ethanol}}{V_{filtered}}) \qquad \ldots\ldots (5)$$

where L is the path length in meters, $V_{ethanol}$ is the volume of ethanol, and $V_{filtered}$ is the filtered volume of
the water samples.
**2.5 HPLC Measurements**
Water samples were filtered through 0.7-µm GF/F filters and immediately frozen in liquid nitrogen for
HPLC analysis using the methods reported by Barlow et al. (1997). The detected pigments along with
their abbreviations are listed in Table 1. Diagnostic biomarker pigments are marked in bold letters (Table
207  1).

**2.6 FIRe Measurements**
An in-situ Fluorescence Induction and Relaxation System (FIRe, Satlantic Inc.) was used to characterize
phytoplankton photosynthetic physiology during the two surveys in GB. The FIRe is based on
illuminating a sample with an intense flash of light to instantaneously saturate the reaction centers of
photosystem II (PSII); under these light conditions, reaction centers do not accept electrons and most of
the absorbed light energy is dissipated as fluorescence. The fundamental parameter measured by FIRe is
fluorescence yield F(t), which is the emitted fluorescence divided by the irradiance intensity (no unit). In
contrast to strong flashes, dark adaption enables all reaction centers of PSII to be open with least
fluorescence emitted, thus, resulting in minimal fluorescence yield ($F_o$). Maximum fluorescence yield ($F_m$)
can be obtained after sufficient irradiation when all reaction centers are closed. Maximum photochemical
efficiency, which quantify the potential of converting light to chemical energy for the PSII reaction
centers (Moore et al., 2006), was calculated as ($F_m$ - $F_o$)/ $F_m$ = $F_v$ / $F_m$. The functional absorption cross
section $\sigma_{PSII}$ ($Å^2$quantum$^{-1}$) measures the capability of reaction centers to absorb light from the ambient
environment. The FIRe was deployed at a slow descent rate, with 12 and 20 vertical profiles obtained
during the first and second surveys, respectively. All measurements were programmed using standard
protocols of single saturating turn-over (ST) flash saturation of PSII (Kolber et al., 1998). Flashes were
generated from highly uniform blue LEDs at 455 nm with approximately 30 nm half-bandwidth. Chl a
fluorescence was stimulated using saturating sequence of 80 1.1 μs flashes applied at 1 μs intervals, 8
sequences were averaged per acquisition, and the fluorescence signal was detected at 668 nm. All data
were processed using standard FIReCom software (Satlantic). In addition, samples of 0.2-μm filtered sea
water at each station were used as 'blank' to remove the background fluorescence signals (Cullen and
Davis, 2003); in this step, the fluorescence from the filtered samples (without phytoplankton) were
subtracted from in-situ fluorescence signals to get more accurate values of $F_v/F_m$.
**2.7 Retrieving Phytoplankton Groups from above-water $R_{rs}$**
A fundamental relationship that links sub-surface remote-sensing reflectance ($r_{rs}$) and the IOPs was
expressed using a quadratic function developed by (Gordon et al., 1988):
$$r_{rs} = g_1 * u(\lambda) + g_2 * u(\lambda)^2; \ u(\lambda) = \frac{b_b}{a_{total}+b_b} \qquad \ldots\ldots (6)$$

where, the parameters $g_1$ (~0.0788) and $g_2$ (~0.2379) are values for turbid estuarine waters (Joshi and
D'Sa, 2018); $r_{rs}$ is the sub-surface remote sensing reflectance that were obtained from above-water remote
sensing reflectance ($R_{rs}$) using (Lee et al., 2002):
$$r_{rs} = \frac{R_{rs}}{0.52+1.7 \times R_{rs}} \qquad \ldots\ldots (7)$$

The total backscattering coefficient $b_b$ is comprised of water ($b_{bw}$) and particulates including both organic
and inorganic particles ($b_{bp}$), while the total absorption coefficients ($a_{total}$) can be further separated  into
four sub-constituents (Roesler and Perry, 1995) as indicated by:
$$b_b = b_{bw} + b_{bp}; \ a_{total} = a_w + a_{phy} + a_{CDOM} + a_{NAP} \qquad \ldots\ldots (8)$$

where $a_w, a_{phy}, a_{CDOM}$, and $a_{NAP}$ represent the absorption coefficients of pure water, phytoplankton, colored
dissolved organic matter and non-algal particles, respectively.
The IOP inversion algorithm for retrieving IOPs from $R_{rs}$ require known spectral shape (eigenvector) of
each component in Eq. (8) to estimate the magnitude (eigenvalues) of each component (Table 2). The
spectral shape can be adjusted by changing the values of slope based on characteristics of the study area.
It is worth noting that a single averaged phytoplankton eigenvector does not provide species information
whereas a set of several species-specific phytoplankton eigenvectors allow estimates of species
composition. IOPs inversion algorithm applied in this study makes use of mass-specific phytoplankton
absorption spectra of 10 groups namely, dinoflagellate, diatom, chlorophyte, cryptophyte, haptophyte,
prochlorophyte, raphidophyte, rhodophyte, red cyanobacteria and blue cyanobacteria; these were obtained
from Dierssen et al. (2006) and Dutkiewicz et al. (2015) as eigen vectors rather than using one average
$a_{phy}(\lambda)$ spectrum. Subsequently, the inversion algorithm iterates repeatedly to minimize the difference
between modeled $R_{rs}$ and in-situ measured $R_{rs}$ ($R_{rs\_insitu}$) until a best fit is achieved while allowing for
alterations of all parameters listed in Table 2 (Chase et al., 2017). The absolute percent errors between
modeled and measured values of $R_{rs}$, $a_{phy}$, $a_{CDOM}$, $a_{NAP}$ and $b_{bp}$ were calculated as:
$$\%error = \left| \frac{X_{modeled}-X_{measured}}{X_{measured}} \right| \times 100 \qquad \ldots\ldots (9)$$

**2.8 Retrieving Pigments from Sentinel 3-OLCI $R_{rs}$**
**2.8.1 Reconstruction of Pigment Absorption Spectrum by Multiple Linear Regression**
Total pigment absorption spectra $a_{pig}(\lambda)$ obtained during both surveys (Eq. 5), were modeled as a third
order function of HPLC measured Chl a (Chl a_HPLC) concentration at each station as (Moisan et al., 2017):
$$a_{pig}(\lambda) = C_3\times(\text{Chl a\_HPLC})^3 + C_2\times(\text{Chl a\_HPLC})^2 + C_1\times\text{Chl a\_HPLC} + C_0 \quad \dots\dots (11)$$
where vector coefficient C=[$C_3$, $C_2$, $C_1$, $C_0$], are wavelength-dependent coefficients that range from 400 to
700 nm at 1 nm interval; these were further applied to Sentinel-3A OLCI Chl a to calculate $a_{pig\_OLCI}$ at
each pixel as:
$$a_{pig\_OLCI}(\lambda) = C_3\times(\text{Chl a\_OLCI})^3 + C_2\times(\text{Chl a\_OLCI})^2 + C_1\times\text{Chl a\_OLCI} + C_0 \quad \dots\dots (12)$$
where Chl a_OLCI is Sentinel-3A OLCI derived Chl a concentration ($259\times224$ pixels); the obtained image
represents the value of $a_{pig\_OLCI}$ at a certain wavelength and 301 images of $a_{pig\_OLCI}$ can be obtained in
the 400-700 nm wavelength range at 1 nm interval.
**2.8.2 Satellite Retrieval of Pigments using Non-Negative Least Square (NNLS) Inversion Model**
The $a_{pig\_OLCI}$ is a mixture of n pigments with known absorption spectra $a_i(\lambda)$, i = 1,2, ... , n at wavelength
$\lambda$ (nm); thus, $a_{pig\_OLCI}(\lambda)$ can be considered as a weighted sum of individual component spectrum
(Thrane et al., 2015) at each image point as:
$$a_{pig\_OLCI}(\lambda) = x_1\times a_1(\lambda) + x_2\times a_2(\lambda) + \cdots x_n\times a_n(\lambda) \quad \dots\dots (13)$$
where $A(\lambda) = [a_1(\lambda), a_2(\lambda), \dots a_n(\lambda)]$ represent the mass-specific spectra of 16 pigments (Chl a, Chl b,
Chl $c_1$, Chl $c_2$, pheophytin-a, pheophytin-b, peridinin, fucoxanthin, neoxanthin, lutein, violaxanthin,
alloxanthin, diadinoxanthin, diatoxanthin, zeaxanthin, and β-carotenoid), which are the in-vitro pigment
absorption spectra over pigment concentrations and can be extracted from supplementary R scripts of
Thrane et al. (2015). The vector coefficient $X = [x_1, x_2, \dots x_n]$ correspond to the concentrations (µg $L^{-1}$) of
these distinct pigments; note that X cannot be negative, therefore, non-negative least squares (NNLS) was
used to guarantee positive solutions of X (Moisan et al., 2013; Thrane et al., 2015). Eq. 13 can be further
expressed as:

$$\begin{bmatrix} a_{pig}(400)_{OLCI} \\ a_{pig}(401)_{OLCI} \\ \vdots \\ a_{pig}(700)_{OLCI} \end{bmatrix} = \begin{bmatrix} x_1 \\ x_2 \\ \vdots \\ x_n \end{bmatrix} \times \begin{bmatrix} a_1(400), a_2(400), \dots a_n(400) \\ a_1(401), a_2(401), \dots a_n(401) \\ \vdots \\ a_1(700), a_2(700), \dots a_n(700) \end{bmatrix} \quad \dots\dots (14)$$
**2.9 Processing Approach**
Sentinel 3A-OLCI pigment maps were generated using the processing pathway 1 (Fig. 2) that includes the
following: 1) developing empirical relationships between HPLC-measured Chl a and $R_{rs\_insitu}$ band ratio
for Sentinel 3A-OLCI band 9 (673 nm) and band 11 (709 nm) to generate Sentinel 3A-OLCI Chl a maps,
and 2) converting Chl a concentration to $a_{pig\_OLCI}(\lambda)$ maps, and subsequently decomposing $a_{pig\_OLCI}(\lambda)$
into individual pigment spectra to generate phytoplankton pigment maps for GB. In processing pathway 2,
phytoplankton taxonomic composition at each sampling station was obtained from a 10-species IOP
inversion model, which take $R_{rs\_insitu}$ as input and estimates Chl a concentration of each phytoplankton
group (Fig. 2). Finally, CDOM-corrected FIRe measurements of $F_v/F_m$ and $\sigma_{PSII}$ were combined with
phytoplankton taxonomy to assess photosynthetic physiology of different phytoplankton groups.
**3. Results**
**3.1 Phytoplankton Taxonomy and Physiological State from Field Observations**
**3.1.1 Measurements of Above Water Remote Sensing Reflectance**
Above-water remote sensing reflectances ($R_{rs\_insitu}$) from the two surveys (Fig. 3) reflect the influence of
the absorbing and scattering features of water constituents. Low reflectance (~675 nm) caused by Chl a
red light absorption and maximum reflectance at green wavelength (~550 nm) were observed. Obvious
dips at ~625 nm versus reflectance peaks ~650 nm were observed in spectra during both surveys, which
could be attributed to cyanobacteria modulation of the spectra (Hu et al., 2010). The reflectance peak
around 690–700 nm was obvious at most sampling sites except at stations 13 and 14 adjacent to the
nGOM and were likely due to the effect of Chl a fluorescence (Gitelson, 1992; Gilerson et al., 2010). The
peak position at stations with lower Chl a concentration (~5 µg L$^{-1}$) were observed at 690-693 nm;
however, the peaks shifted to longer wavelengths of 705 and 710 nm for station 23 and 19 with extremely
high Chl a concentrations of ~31 and 50 µg L$^{-1}$, respectively (Fig. 3).
**3.1.2 Performance of IOP Inversion Algorithm**
The IOP inversion algorithm was applied to $R_{rs\_insitu}$ data (Fig. 3) obtained during the two surveys in GB.
The mean errors for modeled $a_{CDOM}$, $a_{NAP}$, $a_{phy}$ and $b_{bp\_470}$ at all wavelengths for the 34 stations were
5.86%, 6.83%, 12.19% and 10.79%, respectively (Table 3). A total of 8 phytoplankton groups
(dinoflagellate, diatom, chlorophyte, cryptophyte, haptophyte, prochlorophyte, raphidophyte, and blue
cyanobacteria) were spectrally detected from IOP inversion algorithm. The sum of 8 eigenvalues of $Chl_i$
(Table 2) represents the modeled total Chl a (TChl a$_{\_mod}$) of the whole phytoplankton community. The
TChl a$_{\_mod}$ is better correlated with HPLC-measured total Chl a (TChl a$_{\_HPLC}$) for survey 2 (green circle;
Fig. 4a) with $R^2$ ~ 0.92, compared to survey 1 (red color; Fig. 4a). In addition, the TChl a$_{\_mod}$ appear to be
slightly higher than TChl a$_{\_HPLC}$ for survey 2. The modeled $a_{CDOM}$ ($a_{CDOM\_mod}$) are in close agreement with
spectrophotometrically measured $a_{CDOM}$ at 412 nm (Fig. 4b), with $a_{CDOM}$ obtained on September 29, 2017
much higher than that on October 29-30, 2017. The modeled $b_{bp}$ ($b_{bp\_mod}$) are well correlated with in-situ
$b_{bp}$ ($b_{bp\_insitu}$) at 470 nm (Fig. 4c) with higher $R^2$ (0.81) observed on September 29, 2017. In addition, both
modeled and field-measured $b_{bp}$ showed stronger backscattering at most stations on September 29, 2017
than those on October 29-30, 2017.
The Chl a percentage (%Chl a), which is $Chl_i$/TChl a, were also compared with diagnostic pigment
percentage (%DP), which is specific DP for each phytoplankton group over the sum of DP ($\sum$ DP). The
DP for diatom (fucoxanthin), dinoflagellate (peridinin), cryptophytes (alloxanthin), chlorophyte (Chl b),
haptophyte (19'-hexanoyloxyfucoxanthin), and cyanobacteria (zeaxanthin) referred in Moisan et al. (2017)
were used in this study. The $R^2$ between %Chl a and %DP for different phytoplankton groups range from
0.15 to 0.81 (Fig. 4). The %Chl a of cryptophyte is between 5%-42% and well correlated with
alloxanthin/$\sum$ DP ($R^2$~0.62-0.72; Fig. 4d) for both surveys. In addition, the cryptophyte %Chl a at station
19 and 23 on October 30, 2017 was highest (~40%) in coincidence with the highest value of alloxanthin/
$\sum$ DP (Fig. 4d). Furthermore, relationship between chlorophyte %Chl a and Chl b/$\sum$ DP ($R^2$~0.55; Fig. 4e)
showed that chlorophyte during survey 1 contributed higher fraction to the whole phytoplankton
community compared to survey 2. The %Chl a of cyanobacteria highly correlated with zeaxanthin/$\sum$ DP
with $R^2$ larger than 0.7 (Fig. 4f) for both surveys. Low %Chl a of dinoflagellate in coincidence with low
peridinin/ $\sum$ DP ($R^2$~0.78) were observed at stations along the transect, however, increased contribution
of dinoflagellate appeared adjacent to the entrance during both surveys (Fig. 4g).
### 3.1.3    Variations in Phytoplankton Community Structure
Reconstruction of the phytoplankton absorption coefficients spectra revealed variations in phytoplankton
community structure (Fig. 5) even several weeks after Hurricane Harvey. The modeled $a_{phy}$ spectra
($a_{phy\_mod}$) at stations 6, 13, 17 and 19 (Fig. 5a-f) yielded spatiotemporal differences of phytoplankton
taxonomic composition in GB. The strong absorption peak around 625 nm induced by cyanobacteria was
observed at most of the stations for both modeled results and in-situ measurements (Fig. 5a, c and e)
except at stations adjacent to the entrance (Fig. 5b and d). The $a_{phy\_mod}$ at station 6 was primarily
dominated by group of cyanobacteria (blue line) and chlorophyte (green line) on September 29, 2017 (Fig.
5a); in contrast, the spectrum of chlorophyte contributed very little at station 6 on October 29, 2017
(green line; Fig. 5c). Furthermore, the shape of spectra for samples obtained at station 13 showed strong
dinoflagellate-modulation versus extremely low cyanobacteria contribution during survey 1 (red line; Fig.
5b). However, small-size group like haptophyte and prochlorophyte displayed increasing proportions at
station 13 on October 29, 2017 (Fig. 5d). Station 17 in the East Bay was dominated by cyanobacteria
(blue line; Fig. 5e) and cryptophyte (pink line; Fig. 5e) absorption spectra, whereas, on October 30, 2017,
the main spectral features at station 19 in the upper GB was from cryptophyte (pink line) and chlorophyte
(green line; Fig. 5f).
The corresponding taxa-specific %Chl a derived from IOPs inversion algorithm for the two surveys on
September 29 and October 29-30, 2017 are shown in Figure 6a and 6b, respectively. Cyanobacteria (blue
bars) and chlorophyte (green bars) constituted over 55% of the phytoplankton communities during survey
1 (September 29, 2017; Fig. 6a). In addition, chlorophyte, known to proliferate in freshwater
environments, showed higher fraction than that observed in survey 2 (Fig. 6). Further, chlorophyte
together with diatoms (Fig. 6a) accounted for ~ 60% of TChl a_mod at many stations with a well-mixed
water column (e.g., stations 7, 8 and 9 as inferred from salinity profiles; not shown) on September 29,
2017. Cryptophyte, haptophyte and raphidophyte became a minor component of the community and
accounted in total to ~25% of TChl a_mod (Fig. 6a). Furthermore, contribution by dinoflagellate group to
TChl a_mod was low inside the bay, but showed increasing %Chl a (~30%) in higher salinity waters
adjacent to the nGOM (red color; Fig. 6a). Cyanobacteria (blue color; Fig. 7) exhibited a slightly elevated
percentage during survey 2 (~60 days after hurricane passage, October 29-30, 2017) and were quite
abundant in East Bay (stations 16, 17 and 18) where the water was calm and stratified (as indicated by the
salinity profiles - not shown). In addition, cyanobacteria were not prevailing adjacent to the nGOM
(stations 12, 13 and 14) and close to San Jacinto (stations 19, 20, 21, 23 and 24), where cryptophyte and
chlorophyte showed dominance (Fig. 6b). The %Chl a of chlorophyte obtained at stations along the
Trinity River transect decreased by ~10% on October 29-30, 2017 compared to September 29, 2017.
Small size groups like haptophyte and prochlorophyte increased on October 29-30, 2017 and were more
abundant adjacent to the nGOM, accounting for more than 25% of the TChl a_mod.
### 3.1.4    Environmental Conditions and Physiological State of Phytoplankton Community
The surface salinity presented a pronounced seaward increasing gradient along the transect (stations 3-14)
during both the surveys (Fig. 7a) with primarily lower salinity throughout the bay during survey 1 in
comparison to survey 2, which indicated the freshening impact was still ongoing even 4 weeks after
Hurricane Harvey. The salinity was ~15 at station 16 and decreasing when going further into East Bay
(~10 at stations 17 and 18; Fig. 7a). In upper GB, salinity at stations 19-24 did not vary significantly
(~15), increasing along with the distance away from the San Jacinto River mouth with highest value
(~17.5) at station 24. During both surveys, lowest Chl a (Fig. 7b) were observed adjacent to the nGOM,
and the highest Chl a were closest to the river mouth. The photosynthetically Active Radiation (PAR)
which were calculated from down-welling irradiance (not shown here) decreased significantly with depth,
but surface PAR (Fig. 7c) were similar in magnitude at all stations. Pigment ratios including TChl a/TP
(0.58-0.68), PSC/Chl a (0.07-0.26) and AP/TP (0.34-0.42) were obtained from HPLC measurements and
shown in Figure 7d, e and f, respectively.
The CDOM calibrated and 0-0.5m depth averaged photosynthetic parameters $F_v/F_m$ varied from 0.41 to
0.64 (Fig. 7g),  while $\sigma_{PSII}$ was in the range of 329-668 $Å^2$quantum$^{-1}$(Fig. 7h). The highest $\sigma_{PSII}$ and
lowest $F_v/F_m$ appeared adjacent to the nGOM (stations 12-14). Conversely, values of $F_v/F_m$ at stations 7-9
with a well-mixed water column were high with low values of $\sigma_{PSII}$. Both $F_v/F_m$ and $\sigma_{PSII}$ did not directly
correlate with Chl a (e.g., high Chl a ~51 μg L$^{-1}$ at station 19 corresponded to a relatively low level of
$F_v/F_m$ ~0.45, versus high $\sigma_{PSII}$ ~550 $Å^2$quantum$^{-1}$). However, the stations with high $F_v/F_m$ coincided
with the high fraction of Chl a (Chl a/TP) and low fraction of AP (AP/TP) (Fig. 7d and f). In contrast,
$\sigma_{PSII}$ showed an overall positive relationship with AP/TP, but altered negatively with Chl a/TP during
both surveys. The lowest (highest) value of $\sigma_{PSII}$ ($F_v/F_m$) were observed at station 9 corresponding to the
highest Chl a/TP value (~0.64) on October 29, 2017. The highest AP/TP and PSC/Chl a were obtained
from stations adjacent to the nGOM.

### 401  3.1.5  $F_v/F_m$ and $\sigma_{PSII}$ Taxonomic Signatures

Distinct pigments housed within phytoplankton light-harvesting antennae can strongly influence PSII
light-harvesting capability and the photosynthetic quantum efficiency of phytoplankton (Lutz et al., 2001).
In this study, we observed an inverse relationship ($R^2$~0.63-0.81; Fig.  8a and d) between the $F_v/F_m$ and
$\sigma_{PSII}$, that appeared related to taxonomic signals during surveys 1 and 2 in GB. Stations 1-9 along the
transect were considered as well-mixed group with no dominance by any particular group (black circles;
Fig. 8a-c); stations 10-14 close to the entrance were however, strongly dominated by dinoflagellate and
haptophyte (red symbol; Fig. 8a-c) during survey 1. This well-mixed group displayed low values of $\sigma_{PSII}$
(~390-439 $Å^2$quantum$^{-1}$) and high levels of $F_v/F_m$ (~0.42-0.65), with $F_v/F_m$ approaching 0.65 at station
9 on September 29, 2017 (Fig. 8a). However, enhanced contributions of dinoflagellate and haptophyte
around the entrance corresponded to a decline of $F_v/F_m$ (0.3~0.4) against an increase of  $\sigma_{PSII}$ (500~600
$Å^2$quantum$^{-1}$) during survey 1. Furthermore, samples obtained from survey 2 at stations 1-9, stations
10-14, stations 16-18 and stations 19-24 were considered as well-mixed (black), dinoflagellate-
haptophyte dominated (red), cyanobacteria dominated (blue) and cryptophyte-chlorophyte dominated
(green), respectively. Stations 16-17 dominated by cyanobacteria (blue triangles; Fig. 8d) showed high
level of $F_v/F_m$ (0.5~0.6) and relatively low values of $\sigma_{PSII}$ (300~400 $Å^2$quantum$^{-1}$). The $F_v/F_m$ and
$\sigma_{PSII}$ of cryptophyte-chlorophyte dominated stations showed a moderate level of $F_v/F_m$  (0.4~0.5) and
$\sigma_{PSII}$ (580~680 $Å^2$quantum$^{-1}$ ). More  importantly, tight  positive  relationships  existed  between
measurements of $F_v/F_m$ and Chl a/TP ($R^2$~0.31-0.63; Fig. 8b and e). On the other hand, $\sigma_{PSII}$ were
positively correlated with PSC/Chl a with $R^2$~0.6 (Fig. 8c and f). The PSC/Chl a of cyanobacteria
dominated group (blue symbols), and well mixed group (brown symbols) were relatively low. Highest
PSC/Chl a and lowest Chl a/TP was observed for the dinoflagellate-haptophyte dominated group,
corresponding to the lowest $\sigma_{PSII}$ and highest $F_v/F_m$. In addition, cryptophyte-chlorophyte dominated
group had high levels of PSC/TChl a (~0.18-0.26) and slightly higher Chl a/TP compared to
dinoflagellate-haptophyte dominated group. Overall, well-mixed groups with high proportion of large-
size phytoplankton (e.g., diatoms and chlorophyte) showed higher Chl a/TP along with larger $F_v/F_m$ and
smaller $\sigma_{PSII}$ than those stations with high fraction of dinoflagellate and pico-populations (Fig. 8c and f).

### 428  3.2  Satellite Observations of Phytoplankton Pigments

### 3.2.1 An OLCI Chl a Algorithm and its Validation

Blue to green band ratio algorithms have been widely used to study Chl a in the open ocean and shelf waters (D'Sa et al., 2006; Blondeau-Patissier et al., 2014); however, these bands generally fail in estuarine waters due to strong blue absorption by the high levels of CDOM and suspended particulate matter, especially after flooding events associated with hurricanes (D'Sa et al., 2011; D'Sa et al., 2018; Joshi and D'Sa, 2018). The percentage contribution by CDOM fluorescence (blank) to maximum fluorescence yield ($F_m$) obtained from in-situ FIRe (Fig. 9a) demonstrated that Chl a fluorescence was strongly influenced by high amounts of CDOM fluorescence in GB, especially during the first survey (September 29, 2017), when the bay was under strong floodwater influence (red triangles; Fig. 9a). The CDOM fluorescence signal constituted ~ 25 % in the region adjacent to the nGOM (stations 12-14), between 25%-50% in the upper GB, and up to ~65% in Trinity Bay, which implied that blue and even green bands are highly contaminated by CDOM and might not be the most suitable bands for estimating Chl a in GB. However, an increase in peak height near 700 nm and its shift towards longer wavelength (Fig. 3) can be used as a proxy to estimate Chl a concentration (Gitelson, 1992).

The C2RCC atmospheric-corrected $R_{rs\_OLCI}$ at each of the sampling sites were further compared with $R_{rs\_insitu}$ (Fig. 3) at phytoplankton red absorption (~673 nm) and Chl a fluorescence (~700 nm) bands (Fig. 9b). The C2RCC performed overall better for the second survey on October 29-30, 2017 (green and blue symbols; Fig. 9b) than the first survey on September 29, 2017 (red triangles; Fig. 9b) when stations 1, 3 and 4 (circled triangles; Fig. 3c) adjacent to the Trinity River mouth were included; these stations were the last sampling sites in the afternoon (~4:30 pm) and under somewhat cloudy conditions. The time difference between satellite pass and in-situ measurements, sky conditions and shallow water depth also likely introduced more errors at these locations. The $R^2$ between $R_{rs\_OLCI}$ and $R_{rs\_insitu}$ at red and near infrared (NIR) bands was 0.89 when the data from stations 3 and 4 were excluded, suggesting good usability of these two bands for Chl a empirical algorithms in GB. Thus, the higher the Chl a concentration, the stronger the red light absorption, resulting in higher reflectance at 709 nm; consequently, negative correlations were observed between Red/NIR band ratio and Chl a. The ratio of Red (~673 nm) and NIR (709 nm) reflectance bands from in-situ measurements were overall highly correlated with HPLC-measured Chl a with $R^2$ ~ 0.96, 0.94 and 0.98 on September 29, October 29 and October 30, 2017, respectively (Fig. 9c). The Sentinel-3A OLCI Chl a maps (Fig. 10a-c) were generated for all data based on the relationship between Chl a and the Red and NIR band ratio as:

$$\text{Chl a } (\mu g \, L^{-1}) = 216.38 \times \exp\left(-2.399 \times \frac{R_{rs}\,(673)}{R_{rs}\,(709)}\right) \quad \text{[All data]} \quad \ldots\ldots (15)$$

The OLCI-derived Chl a (Fig. 10a-c) showed a good spatial agreement with Chl a_HPLC (Fig. 10d-f). In addition, a comparison of this algorithm with that of Gilerson et al., 2010 revealed slightly better performance (not shown) inside of GB and especially in the area adjacent to the shelf.

The Chl a concentration on September 29, 2017 was overall higher than on October 29-30, 2017 throughout the bay. East Bay displayed very high Chl a concentration, with highest value (>30 µg L$^{-1}$) observed on September 29, 2017 (Fig. 10a). The narrow shape and shallow topography of East Bay results in relatively higher water residence time (Rayson et al. 2016); thus, the reduced exchange with shelf waters likely lends the East Bay vulnerable to eutrophication. The average Chl a concentration on October 29-30, 2017 were ~ 15 µg L$^{-1}$ along the transect (station 1-11) and ~4-6 µg/L (station 12-14) close the entrance of GB. In addition, Chl a adjacent to San Jacinto River mouth (>16 µg L$^{-1}$) was higher than that in Trinity Bay, which might suggest that San Jacinto inflow had higher nutrient concentrations than Trinity as also previously reported (Quigg et al., 2010). Furthermore, the OLCI-Chl a maps on October 29 and 30, 2017 showed extremely high Chl a concentration in a narrow area adjacent to the San Jacinto River mouth, with Chl a approaching ~ 40 µg L$^{-1}$ at station 19 (Fig. 10c).

### 3.2.2 Long-term Chl a Observations in Comparison with Hurricane Harvey Event

OLCI-derived Chl a maps between August, 2016-November, 2017 (Fig. 11$a_1$-$a_{15}$) and time series of
averaged Chl a in the areas of Trinity Bay, East Bay and adjacent to the nGOM (Fig 11b) revealed
regionally different responses to freshwater discharge from the San Jacinto and the Trinity Rivers (Fig.
11b). Due to the relatively much higher discharge from the Trinity River, the spatial distribution of Chl a
in the bay (Fig. 11) generally indicates its greater influence than the San Jacinto River. During the winter
and spring of 2017, phytoplankton Chl a peaks of ~32 µgL$^{-1}$ were observed in Trinity Bay (Fig. 11b) after
high inflows from both rivers (Fig. 11$a_5$-$a_8$) and then decreased to ~20 µgL$^{-1}$ in summer (July and August,
2017). Generally, Chl a showed overall lower value (~10 µgL$^{-1}$) between September-December, 2016
compared to 2017 in the absence of significant meteorological and hydrological events (Fig. 11$a_1$-$a_4$).
However, with the East Bay less directly affected by river discharge, Chl a levels remained fairly constant
in the range of ~18-24 µgL$^{-1}$ before the hurricane. In contrast, extremely high river discharge (~3300 m$^3$s$^{-1}$)
induced by Hurricane Harvey in late August, 2017, elevated Chl a in both the Trinity and East Bay to
higher levels as observed on September 14, 2017 (~30-35 µgL$^{-1}$; Fig. 11$a_{11}$) compared to the mean state
of fall season in 2016. Chl a then continuously decreased through September and October, 2017 in Trinity
and East Bay, and were relatively low (≤10 µgL$^{-1}$) in November, 2017 under no additional pulses of river
discharge. Chl a adjacent to the entrance of GB which exhibited much lower values year round than that
of the Trinity and East Bay, also showed slight positive response to the enhanced river discharge and the
hurricane-induced flooding events. In addition, Chl a always displayed low values along the Houston
Ship Channel.
### 3.2.3 Reconstruction of Total Pigment Absorption Spectra from OLCI-derived Chl a
The reconstructed $a_{pig}(\lambda)$ based on the third order function of Chl a$_{HPLC}$ (gray lines; Fig. 12a and b)
agreed well with the spectrophotometrically measured $a_{pig}(\lambda)$ (black lines; Fig. 12a and b) during both
surveys ($R^2$=0.86; Fig. 12c). The $R^2$ for modeled versus measured $a_{pig}(\lambda)$ are between 0.76 and 1.00 from
400 to 700 nm with averaged $R^2$ of whole spectra reaching ~ 0.82 on September 29, 2017 and ~0.89 on
October 29-30, 2017, respectively. The vector coefficients $C = [C_3, C_2, C_1, C_0]$ obtained from Eq. (11)
were further applied to Eq. (12) to generate $a_{pig\_OLCI}(\lambda)$ based on OLCI-derived Chl a images on July 06
(Fig. 11$a_9$), September 29 (Fig. 10a), October 29-30 (Fig. 10b-c), and Nov 25 (Fig. 11$a_{15}$), 2017,
respectively; these contained 259×224 pixels in each image. The $a_{pig\_OLCI}(\lambda)$ at each pixel was retrieved
at 1 nm interval, and thus 301 images of $a_{pig\_OLCI}(\lambda)$ representing each wavelength were obtained over GB.
### 3.2.4 Accuracy of phytoplankton pigment retrievals from Sentinel 3A-OLCI
The reconstructed $a_{pig\_OLCI}(\lambda)$ was spectrally decomposed into 16 individual pigment spectra at each pixel
based on Eq. (14). A comparison of all data between HPLC-measured pigments and NNLS algorithm
inverted pigments showed that $R^2$ ranged from a low of 0.40 for diatoxanthin to 0.96 for Chl a and RMSE
was in the range of 0.103-0.584 (Table 4). The NNLS-modeled Chl a also correlated well with OLCI-
derived Chl a ($R^2$=0.98; Fig. 13a), with each exhibiting similar quantitative and spatial patterns. For the
other 15 simultaneously simulated pigments, $R^2$ of only 7 pigments were greater than 0.650 (Table 4). In
addition, the resulting RMSE were less than 0.3 for most of pigments, except zeaxanthin, violaxanthin,
diatoxanthin and diadinoxanthin. Further, for those pigments with relatively lower RMSE, their slopes
were very close to 1 and y-intercepts approached to 0. Five NNLS-derived versus HPLC measured
diagnostic pigments including alloxanthin, Chl b, zeaxanthin, fucoxanthin and peridinin are shown in
Figure 13. The $R^2$ between NNLS-derived and HPLC-measured pigments for surveys 1 and 2 was highest
for alloxanthin (0.91; Fig. 13b). For the other pigments $R^2$ was 0.854 for Chl b (Fig. 13c), 0.689 for
zeaxanthin (Fig. 13d), 0.645 for fucoxanthin (Fig. 13e) and 0.566 for peridinin (Fig. 13f), respectively.
### 3.3 Spatiotemporal Variations of Diagnostic Pigments
Flooding due to Hurricane Harvey not only enhanced Chl a, but also affected the phytoplankton pigments
composition. NNLS-retrieved pigment maps for July, September, October and November, 2017 including
those of alloxanthin, chl b, zeaxanthin, fucoxanthin and peridinin (Fig. 14) showed different levels of
variations before and after the hurricane event. Alloxanthin, which is unique to cryptophytes (Wright and
Jeffrey, 2006) exhibited same spatial distribution patterns (Fig. 14$a_1$-$e_1$) with Chl a. Alloxanthin was
especially low (~ 0.5 µg L$^{-1}$, Fig. 14$a_1$) in the major basin area on July 06, 2017 before the hurricane and
slightly elevated (~0.7 µg L$^{-1}$, Fig. 14$b_1$) in September and October, 2017 after the hurricane passage.
Furthermore, extremely high alloxanthin (~ 3.5 µg L$^{-1}$, Fig. 14$c_1$-$d_1$) was observed adjacent to San Jacinto
River mouth on October 29-30, 2017, which coincided with the high %Chl a of cryptophyte at stations 19
and 23 (Fig. 6b). The bloom with high concentration of alloxanthin on October 29, 2017 (~3.5 µg L$^{-1}$; Fig.
14$c_1$) then extended to a broader area on October 30, 2017 (Fig. 14$d_1$).
Chl b is abundant in the group of chlorophyte (green algae) (Hirata et al., 2011) and the spatial
distributions of Chl b (Fig. 14$a_2$-$e_2$) also showed strong correlations with Chl a on July 06, 2017,
September 29, October 29-30 and November 25, 2017. The NNLS-derived Chl b exhibited overall low
values (~0.5-2 µg L$^{-1}$; Fig. 14$a_2$) before the hurricane and showed obvious elevation throughout the bay
after the hurricane and eventually decreasing to pre-hurricane level by November 25, 2017. Furthermore,
Chl b concentrations observed on September 29, 2017 were higher than that on October 29-30, 2017,
which corresponded to a decline of chlorophyte percentage derived from the IOP inversion algorithm (Fig.
6). More importantly, images obtained from IFCB at the entrance to GB also detected freshwater species
Chlorophyte (*Pediastrum duplex;* Fig. 14g) on September 29, 2017. However, this species was rarely
observed in IFCB images for the other dates (Fig. 14$a_1$ and Fig. 14$c_1$-$e_2$). In addition, Chl b concentrations
approached ~2.8 µg L$^{-1}$ in the bloom area and the corresponding green discoloration of water was also
observed during the field survey on October 30, 2017.
Zeaxanthin is known as taxa-specific pigment for prokaryotes (cyanobacteria) (Moisan et al., 2017;
Dorado et al., 2015) and NNLS-derived zeaxanthin maps (Fig. 14$a_3$-$e_3$) displayed significantly different
patterns with Chl a, exhibiting low concentrations in the areas where the Chl a were high. For example,
zeaxanthin was especially low in the bloom area on October 29-30, 2017, which agreed well with
low %Chl a of cyanobacteria at stations 19 and 23 (Fig. 6), thus indicating that this localized algal bloom
event was not associated with cyanobacteria. In addition, zeaxanthin was high ~ 3.0 µg L$^{-1}$ (Fig. 14$a_3$) in
both GB and shelf waters on July 06, 2017 before the hurricane event. Later, zeaxanthin increased slightly
on September 29, 2017 (Fig. 14$b_3$) with IFCB data detecting $N_2$-fixing cyanobacteria (*Anabaena* spp.; Fig.
14g) and remained elevated on October 29-30, 2017 (Fig. 14$b_3$-$c_3$). Zeaxanthin eventually decreased to
very low values (~ 1.2 µg L$^{-1}$; Fig. 14$e_3$) on November 25, 2017.
Fucoxanthin is a major carotenoid found in diatoms (Hirata et al., 2011; Moisan et al. 2017) and the
NNLS-derived fucoxanthin maps (Fig. 14$a_4$-$e_4$) showed highly similar distribution patterns with Chl a.
Maps of fucoxanthin showed low concentrations on July 06, 2017 (~1.5 µg L$^{-1}$; Fig. 11$a_4$), and displayed
a large increase on September 29, 2017 (~1.6-3.0 µg L$^{-1}$; Fig. 11$b_4$). Diatom group detected from IFCB
were dominated by marine species before the hurricane, but subsequently shifted to freshwater species
(e.g., *Pleurosigma*; Fig. 14g) and then back to marine species after October, 2017. Overall, fucoxanthin
concentrations in GB were relatively higher during survey 1, which corresponded to the higher %Chl a of
diatom (Fig. 6) compared to survey 2. Although, fucoxanthin decreased to low values on November 25,
2017 (~1.6 µg L$^{-1}$; Fig. 11$e_4$), it accounted for higher fraction of phytoplankton diagnostic pigments
compared to other dates in July, September and October, 2017.
Peridinin, a primary bio-marker pigment for certain dinoflagellates (Örnólfsdóttir et al., 2003), also
displayed significantly distinct patterns in comparison to Chl a (Fig. 14$a_5$-$e_5$). On July 06, 2017, peridinin
was ~0.24-0.36 µg L$^{-1}$, accounting for a high proportion of the diagnostic pigments; meanwhile, diversity
of marine dinoflagellate species observed from IFCB at this time was also high (Fig. 14f). However,
peridinin decreased (~0.001-0.05 µg L$^{-1}$) after the hurricane, with freshwater dinoflagellate species
(*Ceratium hirundinella*; Fig. 14g) detected from IFCB on September 29, 2017. In addition, maps of
peridinin during both surveys (Fig. 14b$_5$-d$_5$) presented higher concentration (~0.3 µg L$^{-1}$) in higher salinity
waters adjacent to the bay entrance, which agreed well with the increasing fraction of dinoflagellate at
stations 10-14 detected from IOP inversion model (Fig. 6). In contrast, peridinin showed low
concentrations in both GB and shelf waters on November 25, 2017 (Fig. 14e$_5$), with dinoflagellate species
rarely observed from IFCB (Fig. 14*l*).

## 575  4    Discussion

### 576  4.1 Performance of the Semi-Analytical IOP Inversion Algorithm

The residuals between R$_{rs\_insitu}$ and R$_{rs\_mod}$ on September 29 and October 29-30, 2017, are negative in the
blue (400-450 nm) and red (610-630 nm) spectral range at most stations, whilst keeping positive ~700 nm,
which could be attributed to a number of factors. First, the underestimation near 700 nm by the IOP
inversion model is possibly induced by the absence of a fluorescence component in the IOP inversion
model; thus, R$_{rs\_insitu}$ containing fluorescence signals were generally higher than R$_{rs\_mod}$ near 700 nm.
Second, in the range of 610-630 nm, the absorption was overestimated at most of the stations; in this
spectral range, the shape of spectra was strongly modulated by cyanobacteria absorption. Thus this
overestimation at ~620 nm is likely introduced by the input absorption spectrum (eigenvector) for
cyanobacteria since all of input a$^*_{phi}$ ($\lambda$) are general absorption spectral shapes for different phytoplankton
groups. However, the spectra of a$^*_{phi}$($\lambda$) can vary in magnitude and shape associated with package effects
under different environmental conditions (e.g. nutrient, light and temperature) even for the same species
(Bricaud et al., 2004). More detailed absorption spectra of phytoplankton under different conditions (e.g.,
high/low light and nutrients) could improve the performance of the IOP algorithm. Furthermore, the role
of scattering might be another key factor to explain differences between R$_{rs\_insitu}$ and R$_{rs\_mod}$ for the whole
spectra. The quantity and composition of suspended materials including phytoplankton, sediment and
minerals will collectively determine b$_{bp}$($\lambda$) in both shape and magnitude. However, the input eigenvector
of b$_{bp}$($\lambda$) in the present study was not divided into detailed sub-constituents and was a sum spectrum based
on a power law function (Table 2). In reality, b$_{bp}$($\lambda$) spectra are not smooth and regular, and thus, the b$_{bp}$($\lambda$)
value of phytoplankton and sediment might introduce errors to the whole spectrum due to their own
scattering characteristics.

### 597  4.2 Distributions of NNLS-Retrieved Phytoplankton Pigments from Sentinel-3A OLCI

The NNLS-inversion algorithm showed relatively higher R$^2$ for those pigments that better correlated with
HPLC-measured Chl a (e.g., Chl b, and alloxanthin), which was reasonably consistent with Moisan et al.,
2017; this outcome could potentially be attributed to the fact that the NNLS pigment inversion algorithm
was developed based on the relationship between HPLC-measured Chl a and spectrophotometer-
measured a$_{pig}$($\lambda$). For instance, pigments that were relatively poorly correlated with HPLC-measured Chl a,
such as fucoxanthin, diatoxanthin and diadinoxanthin on October 29-30, 2017, the OLCI-derived
concentrations in cryptophyte-chlorophyte algal bloom area showed higher concentrations than those of
HPLC measurements (e.g., values in gray circle; Fig. 13e), thus, resulting in lower R$^2$. However, in
previous studies (Moisan et al., 2017; Pan et al., 2010), satellite-derived fucoxanthin appeared better
correlated with HPLC-measured fucoxanthin compared to this study; it reasons that, fucoxanthin,
generally one of the most abundant diatom biomarker pigments in coastal waters and along with its long-
term measurements in their study area (United States northeast coast), agreed very well with Chl a. In
contrast, the cryptophyte-chlorophyte algal bloom area with extremely high Chl a appeared to disturb the
correlations between Chl a and fucoxanthin in this study. Also, pigments with apparent high values in
algal bloom areas, such as Chl b, Chl c, alloxanthin, lutein, showed higher $R^2$ with RMSE less than 0.3.
Thus, in-situ measurements of Chl a and $a_{pig}(\lambda)$ in waters with stronger gradients in magnitude and greater
variations in phytoplankton community structures could potentially increase the challenge of applying
NNLS-inversion algorithms in optically-complex estuarine waters. Further, the highly dynamic estuarine
environment could as well contribute to additional uncertainties in the validation of inverted pigments due
to variations such as turbulence, turbidity or light field that are likely to occur during the time interval
between in-situ and Sentinel 3-OLCI (~4 hours) observations. HPLC measurements also cannot detect
extremely low pigment concentrations; for example, HPLC-measured peridinin were 0.001 $\mu gL^{-1}$ at
several stations, however, OLCI-derived peridinin showed higher and variable values at these stations
(data in gray circles; Fig. 13f); this could thus increase the RMSE of the NNLS-inverted peridinin. It was
also found that the slope of all pigments were smaller than 1, which demonstrate that NNLS-inverted
pigments were relatively smaller than HPLC measurements, especially for those stations located in the
algal bloom area; this could most likely be attributed to the underestimation of Chl a values by the
Sentinel 3-OLCI empirical algorithms in the algal bloom area. Algal bloom dominated by cryptophyte
group, which is also known to cause red tides worldwide, to some degree, could increase red reflectance
and thus increase ratio values of Red/NIR and decrease estimated Chl a values. Therefore, reliable
estimates of satellite Chl a is crucial for the accuracy of retrieved pigments. The goal of the empirical Chl
a algorithm for Sentinel 3A-OLCI is to obtain more accurate estimation of surface Chl a concentration,
which is better for retrieving other accessory pigments. However, the primary limitation of Chl a
empirical algorithms in this study was that the derived relationships between Red/NIR and Chl a in GB
may only be valid within a specific time period due to temporally-limited field observations versus highly
dynamic estuarine environments. Therefore, a Chl a empirical algorithm that is more broadly applicable
over a longer time period will largely improve the accuracy of retrieved pigments over a series of remote
sensing images and can be more useful for spatiotemporal studies of phytoplankton functional diversity.
More importantly, the highly similar absorption spectra of many carotenoids are another key issue
limiting the accuracy of spectral decomposition techniques. Although the 16 input pigment spectra used
in this study were selected from (Thrane et al., 2015), which were correctly identified from unknown
phytoplankton community structure with low error rate reported from Monte Carlo tests, the potential
effects of aliasing spectra of  some pigment pairs (e.g., fucoxanthin vs peridinin, diadinoxanthin vs lutein,
β-Carotene vs zeaxanthin) could still be a factor. Thus, the reported errors/$R^2$ for retrieved total
carotenoids in Thrane et al., 2015 were apparently lower/higher than those of modeled total chlorophylls,
which showed consistency with this study. Although the predicted pigments showed a range of $R^2$ and
RMSE with known uncertainties, all are within the acceptable range and could be useful for studying the
spatiotemporal responses of PFTs to environmental variations, especially in such optically-complex
estuaries.
The derived maps of phytoplankton diagnostic pigments appeared to be reasonably correlated with
HPLC-measured diagnostic pigments and showed overall agreement with extracted phytoplankton
taxonomic compositions detected from the IOP inversion algorithm. The retrieved diatom-specific
fucoxanthin maps however, showed high concentrations compared to other pigments adjacent to the
entrance (Fig. 13b$_4$ and c$_4$), which contradicted with diatom %Chl a calculated from IOP inversion
algorithm that Chl a fraction of diatom was relatively uniform at stations 12-14 (Fig. 6b). (Nair et al.,
2008) concluded that fucoxanthin can occur in other phytoplankton types (e.g. raphidophyte and
haptophyte). Fucoxanthin and/or fucoxanthin derivatives such as 19′-hexanoyloxyfucoxanthin can also
replace peridinin as the major carotenoid in some dinoflagellates (e.g., Karenia brevis; Jeffrey and Vest,
1997). The elevated contributions from groups of dinoflagellate, haptophyte and prochlorophyte adjacent
to the entrance (stations 10-14; Fig. 6b) along with high concentrations of fucoxanthin likely suggest the
presence of elevated fractions of haptophyte and dinoflagellate, and further implies that fucoxanthin is an
ambiguous marker pigment for diatoms. This could also explain the poor correlation between
inverted %Chl a and %DP observed for the groups of diatom and haptophyte (Fig. 4g and l). These results
also further suggest the inherent limitations of using DP-type comparison between major biomarker
pigments and phytoplankton groups because the major assumption for DP-type methods is that diagnostic
pigment of distinct phytoplankton groups are uncorrelated to each other. This assumption is invalid in that
concentrations of major biomarker pigments are significantly correlated with each other and also may
vary in time and space under some external environmental stress (e.g., temperature, salinity, mixing, light
and nutrient) (Latasa and Bidigare, 1998).
**4.3 Response of Phytoplankton Taxa to Environmental Conditions**
Previous studies showed diatoms to be the most abundant taxa in GB, and tend to be more dominant
during winter/spring, corresponding to periods of high fresh water discharge and nutrient-replete
conditions (Dorado et al., 2015; Örnólfsdóttir et al., 2004a); transition from chain-forming diatoms such
as *Chaetoceros* and rod-like diatoms pre-flood to small cells, such as *Thalassiosira* and small pennate
diatoms were generally observed during high river discharge periods (Anglès et al., 2015; Lee, 2017). In
contrast, cyanobacteria were the most abundant species during the warmer months (Jun-Aug) when river
discharge was relatively low (Örnólfsdóttir et al., 2004b). Further, phytoplankton groups in GB responded
differentially both taxonomically and spatially to the freshening events due to their contrasting nutrient
requirements and specific growth characteristics. For instance, most phytoplankton taxa (e.g., diatom,
chlorophyte and cryptophyte) can be positively stimulated by fresh inflows due to their relatively rapid
growth rate (Paerl et al., 2003); however, Roelke et al. (2013) also documented that cyanobacteria and
haptophytes in the upper GB were not sensitive to nutrient-rich waters from both rivers, due to the extra
nutrients obtained from $N_2$-fixation abilities and mixotrophic characteristics, respectively. In the lower
part of GB, dinoflagellates and cyanobacteria are known to be more dominant during the low river
discharge due to their preference for higher phosphorus (P) compared to some other groups, and to low
turbulence (Lee, 2017) and thus, generally inversely related to the fresh inflows (Lee, 2017; Roelke et al.,
684 2013).

Perturbations following Hurricane Harvey affected the phytoplankton taxonomic composition with
alterations in phytoplankton community structure observed as the GB system transitioned from marine to
freshwater then to marine system (Figs. 6 and 14). Higher fraction of zeaxanthin and peridinin and the
presence of large and slow-growing marine dinoflagellates detected from IFCB pre-hurricane (July 06,
2017) indicate that both cyanobacteria and dinoflagellates were the main groups of phytoplankton
community during summer, and likely associated with warmer temperature and lower river flow (Lee,
2017). Later, massive Chl a observed in September, 2017 and the decline of Chl a to background state in
October, 2017, were likely associated with the hurricane-induced high river discharge and the resulting
variations in nutrient concentration and composition. Higher fractions of diatom and chlorophyte
accompanied by increasing fucoxanthin and Chl b on September 29, 2017, to some extent agreed well
with measurements of Steichen et al., 2018 two weeks following Hurricane Harvey that freshwater
species (diatom, green algae and cyanobacteria) appeared immediately following the flooding event.
Greater abundance of diatom and chlorophyte during survey 1 in comparison to survey 2 were likely due
to their rapid growth rates, enhanced nutrient uptake rates, and tolerance of low salinity and high
turbulence under high nutrient loading conditions following the freshwater inflows (Roy et al., 2013;
Santschi, 1995). Therefore, it is not surprising that Chl b concentrations showed very low values in July
and November, 2017, when river discharge was correspondingly low. Cyanobacteria, which normally
prefer low salinity conditions, also showed specific responses to this flood event. On September 29, 2017,
zeaxanthin slightly increased compared to summer season in July, 2017. The decline of diatoms and
chlorophyte versus slightly increased cyanobacteria observed on October 29-30, 2017, could be attributed
to the relatively slow growth rates of cyanobacteria compared to that of chlorophytes and diatom (Paerl et
al., 2003); cyanobacteria appeared to have lagged behind these groups in terms of responding to enhanced
freshwater discharge when longer residence times were again restored. In contrast, the presence of green
algae and cyanobacteria could as well as be explained by the clarity and turbidity gradient of water. Quigg
et al. (2010) reported that when turbidity was relatively high, chlorophyte dominated over cyanobacteria
with biomass ratio of chlorophyte/cyanobacteria greater than 2, which supported our observations that
chlorophyte dropped off whilst cyanobacteria increased during survey 2 on October 29-30, 2017. In
addition, highest cyanobacteria percentage in East Bay suggest that calm and stratified waters may
accelerate cyanobacteria growth as the buoyancy regulation mechanism of cyanobacteria is possibly
restricted by the water mixing (Roy et al., 2013). Peridinin, which initially decreased in September and
then increased in the lower GB on October 29-30, 2017, suggest that dinoflagellates showed overall
preference for high-salinity waters. Furthermore, previous IFCB observations from Biological and
Chemical Oceanography Data Management Office (BCO-DMO) showed that algal blooms after
hurricanes in the nGOM were initially dominated by diatoms, and subsequently transitioned to blooms of
dinoflagellates, likely associated with nutrient ratios and chemical forms of nutrient supplied by the flood
waters and rainfall (Heisler et al., 2008). In addition, high concentrations of peridinin observed along the
Houston Ship Channel, might provide evidence that the ballast water addition from shipping vessels
likely promote harmful species of dinoflagellates (Steichen et al., 2015). Finally, low concentrations of all
pigments on November 25, 2017 with relatively higher fraction of fucoxanthin compared to previous
dates (Fig. 14), indicate the major role of marine diatoms at that time and further confirms that diatoms
can be found under a wide range of inflows in GB.
The localized cryptophyte-chlorophyte bloom that occurred ~60d after Hurricane Harvey on October 29-
30, 2017, was captured by both satellite and in-situ measurements. This bloom might not be associated
with the flooding events of Hurricane Harvey, and could be linked to nutrient-rich runoff flowing into GB,
reflecting sensitivity and rapid response of phytoplankton community to nutrient input in GB. In shallow
and turbid estuaries, human activities are altering the environment and causing phytoplankton changes in
diversity and biomass to occur more frequently. Dugdale et al. (2012) reported that variations of
phytoplankton community in San Francisco estuary could be attributed to anthropogenically-elevated
concentration of ammonium, which restrain the uptake of nitrate, thus reducing the growth and
reproduction of larger diatoms and shifting towards smaller species (e.g., cryptophyte and green
flagellate). Furthermore, 'pink oyster' events related to alloxanthin of cryptophyte in GB occurred more
frequently from September through October in recent years (Paerl et al., 2003). The eastern side of
Houston Channel in mid bay region was reported as the area most heavily impacted by the intense 'pink
oyster' events. Previous studies and present observations both suggest that this cryptophyte-chlorophyte
dominated bloom could be promoted by the nutrient-driven eutrophication from Houston Ship Channel,
urbanization and industrialization along the upper San Jacinto River complex.
**4.4 Photo-Physiological State of Natural Phytoplankton Community**
In this study, the CDOM-corrected $F_v/F_m$ and $\sigma_{PSII}$ likely represented a composite of both phytoplankton
taxonomy and physiological stress (e.g., nutrient and mixing). Typically, lowest N and P concentrations
were measured closest to the nGOM (Quigg et al., 2009). Phytoplankton community living close to
nGOM were usually in poor nutrient conditions and would be expected to maximize their light harvesting
(increase in $\sigma_{PSII}$) due to nutrient stress. Simultaneously, phytoplankton cells might experience a decline
of functional proportion of reaction centers of PSII (RCII), which means decrease in $F_v/F_m$. The observed
low levels of $F_v/F_m$ and Chl a/TP versus high values of $\sigma_{PSII}$ and AP/TP adjacent to the nGOM showed
agreement with previous studies that the fraction of carotenoids to be higher for nutrient-poor cultures
(Schitüter et al., 1997; Holmboe et al., 1999). In contrast, phytoplankton in well-mixed waters (station 7-9)
might experience abundant nutrients due to the resuspension of cyclonic gyre around Smith Points; as
such, their photosynthetic machinery were likely healthier. Aiken et al. (2004) documented that the Chl
a/TP ratio was relatively higher when plants were in good growing conditions, which is similar to the
observations in this study that phytoplankton have higher fraction of Chl a accompanying higher rate of
photosynthetic efficiency ($F_v/F_m$) under nutrient replete conditions. Overall, the spatial pattern of $F_v/F_m$
and $\sigma_{PSII}$ in GB could be mainly attributed to physiological stress of nutrient and hydrodynamic
conditions since the light availability (PAR) during the sampling period did not spatially vary
significantly at the surface. Furthermore, FIRe measurements ($F_v/F_m$ and $\sigma_{PSII}$) also presented a
taxonomic signal super-imposed upon environmental factors. Each cluster with different dominant taxa
(well mixed group, chlorophyte & cryptophyte, cyanobacteria, and dinoflagellate & haptophyte)
displayed different physiological characteristics. The taxonomic sequence of eukaryotic groups from high
$F_v/F_m$, low $\sigma_{PSII}$ to low $F_v/F_m$, high $\sigma_{PSII}$ in the present observations showed potential effects of
phytoplankton cell size corresponding to diatoms, chlorophyte, and cryptophyte, dinoflagellate and
haptophyte. The prokaryote (cyanobacteria) had relatively high values of $F_v/F_m$ and low values of $\sigma_{PSII}$;
this agreed with $F_v/F_m$ for some species of nitrogen-fixing cyanobacteria that can range from 0.6 to 0.65
(Berman-Frank et al., 2007). Yet, it is difficult to separate the contributions from environmental factors
and taxonomic variations to the changes of FIRe fluorescence signals since all these parameters are inter-
related. Different phytoplankton groups/sizes will display distinct physiological traits ($F_v/F_m$ and $\sigma_{PSII}$)
when experiencing considerable environmental pressures. Thus, effects of physiological stress on
$F_v/F_m$ and $\sigma_{PSII}$ variations for natural samples can only be determined when taxonomic composition can
be excluded as a contributor (Suggett et al., 2009).
**5   Conclusions**
Field measurements (salinity, pigments, optical properties and physiological parameters) and ocean color
observations from Sentinel-3A OLCI were used to study the effects of extreme flooding associated with
Hurricane Harvey on the phytoplankton community structures, pigment distributions and their
physiological state in GB. Flooding effects made the entire GB transition from saline to freshwater then
back to a more marine influenced system. The band ratio (Red/NIR) of $R_{rs\_insitu}$ were negatively correlated
with HPLC-measured Chl a in an exponential relationship ($R^2 > 0.93$). The satellite-retrieved Chl a maps
yielded much higher Chl a concentrations on September 29, 2017 compared to October 29-30, 2017 with
lowest Chl a observed adjacent to the shelf waters. Phytoplankton taxonomic composition was further
retrieved from $R_{rs\_insitu}$ using a 10-species IOP inversion algorithm. Phytoplankton community generally
dominated by estuarine marine diatoms/dinoflagellates before flood events, was altered to freshwater
species of diatom, green algae (chlorophyte) and cyanobacteria during survey 1. It also showed an
increase of small-size species including cryptophyte, haptophyte, prochlorophyte and cyanobacteria
accompanied by a decline of chlorophyte and diatoms during survey 2.
Phytoplankton diagnostic pigments retrieved using an NNLS inversion model based on Sentinel-3A
OLCI Chl a maps also confirmed spatiotemporal variations of phytoplankton taxonomy. The NNLS-
retrieved diagnostic pigment maps showed overall spatiotemporal agreement with HPLC measurements
with $R^2$ ranging from 0.40 (diatoxanthin) to 0.96 (Chl a) during both surveys. Alloxanthin, Chl b, and
fucoxanthin which exhibited similar patterns with Chl a, showed different levels of increase after
Hurricane Harvey. In contrast, NNLS-derived zeaxanthin and peridinin presented significantly low values
in the area where Chl a concentrations were high. Further, maps of zeaxanthin and peridinin displayed
relatively higher fraction on July 06, 2017 before the hurricane compared to other diagnostic pigments.
However, peridinin decreased post-hurricane on September 29, 2017 and then increased a bit on October
29-30, 2017. Concentrations of Chl a and all biomarker pigments eventually decreased to low levels in
November, 2017 when GB returned to its typical environmental state.
Finally, the retrieved phytoplankton taxonomic compositions from IOP inversion algorithm were linked
with FIRe-measured photosynthetic parameters ($F_v/F_m$ and $\sigma_{PSII}$) to assess the effects of physiological
stress and taxonomic contributions on phytoplankton photosynthetic performance. An inverse relationship
between the $F_v/F_m$ and $\sigma_{PSII}$ were observed during both surveys. Phytoplankton community in well-mixed
waters (around Smith Point) showed high $F_v/F_m$ against low $\sigma_{PSII}$; in contrast, the area with poor nutrient

conditions (adjacent to the shelf waters), showed low $F_v/F_m$ and elevated $\sigma_{PSII}$. Taxonomic signatures of $F_v/F_m$ and $\sigma_{PSII}$ revealed diverse physiological characteristics with dinoflagellate-haptophyte group showing the lowest $F_v/F_m$ versus the highest $\sigma_{PSII}$, whereas prokaryote of cyanobacteria-dominated group showed high values of $F_v/F_m$ and low values of $\sigma_{PSII}$. Overall, this study using field and ocean color data combined with inversion algorithms provided novel insights on phytoplankton response to an extreme flood perturbation in a turbid estuarine environment based on taxonomy, pigment composition and physiological state of phytoplankton.

*Data availability.* Data from field measurements are available upon request from the corresponding author.

*Author contributions.* BL and ED conceived and designed the research; BL, ED and IJ collected and processed the data; BL analyzed the data and all authors contributed to writing the paper.

*Competing interests.* The authors declare that they have no conflict of interest.

*Acknowledgements.* The authors thank the European Space Agency (ESA) and the European Organization for Meteorological Satellites (EUMESAT) for providing access to the Sentinel-3 OLCI ocean color data and the Sentinel-3 Toolbox Kit Module (S3TBX) version 5.0.1 in Sentinel Application Platform (SNAP). We also would like to thank the Phytoplankton Dynamics Lab of Texas A&M University at Galveston, for the near-real time microplankton pictures recorded by an Imaging FlowCytobot, which are made available on the web. We are grateful to Bill Gibson from the Coastal Studies Institute for providing logistic support for field operations. EJD acknowledges NASA support through grant No. 80NSSC18K0177.

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

**Table 1**. Pigments information acquired from HPLC samples in Galveston Bay.

| Variable | Primary Pigment (PPig) | Calculation |
|---|---|---|
| | Chlorophylls | |
| [TChl a] | Total chlorophyll a (TChl a) | [Chlide a]+[DVChl a]+[Chl a] |
| [TChl b] | Total chlorophyll b (TChl b) | [DVChl b]+**[Chl b]** |
| [TChl c] | Total chlorophyll c (TChl c) | [Chl c$_1$]+[Chl c$_2$]+[Chl c$_3$] |
| | Carotenoids | |
| [Caro] | Carotenes† | [ββ-Car]+[βε-Car] |
| **[Allo]** | **Alloxanthin** | |
| **[Buta]** | **19'-Butanoyloxyfucoxanthin** | |
| [Diadino] | Diadinoxanthin | |
| [Diato] | Diatoxanthin | |
| **[Fuco]** | **Fucoxanthin** | |
| **[Hexa]** | **19'-Hexanoyloxyfucoxanthin** | |
| **[Peri]** | **Peridinin** | |
| **[Zea]** | **Zeaxanthin** | |
| [Neo] | Neoxanthin | |
| [Lut] | Lutein | |
| [Viola] | Violaxanthin | |
| [Pras] | Prasinoxanthin | |
| [Anthera] | Antheraxanthin | |

Note: (1) [Chl b], [Allo], [Fuco], [Peri], [Zea], [Buta] and [Hexa] are considered as diagnostic pigments for PFTs (Moisan et al., 2017).

| Variable | Pigment Sum | Calculation |
|---|---|---|
| [TChl] | Total Chlorophyll (TChl) | [TChl a]+[TChl b]+[TChl c] |
| [PPC] | Photoprotective Carotenoids (PPC) | [Allo]+[Diadino]+[Diato]+[Zea]+[Caro]+[Viola] |
| [PSC] | Photosynthetic Carotenoids (PSC) | [Buta]+[Fuco]+[Hexa]+[Peri]+[Lut]+[Pras] |
| [PSP] | Photosynthetic Pigments (PSP) | [PSC]+[TChl] |
| [AP] | Total Accessory Pigments (AP) | [PPC]+[PSC]+[TChl b]+[TChl c] |
| [TP] | Total Pigments (TP) | [AP+[TChl a] |
| [DP] | Total Diagnostic Pigments (DP) | [PSC]+[Allo]+[Zea]+[T Chl b] |


**Table 2.** Parameters and eigenvectors used in the semi-analytical inversion algorithm.

| Parameter | Equation | Slope | Eigenvalue |
|---|---|---|---|
| $a_{CDOM}(\lambda)$ | $a_{CDOM}(\lambda) = M_{CDOM} \times exp^{-S_{CDOM} \times (\lambda - \lambda_0)}$; $\lambda_0 = 443$ | $S_{CDOM}$ | $M_{CDOM}$ |
| $a_{NAP}(\lambda)$ | $a_{NAP}(\lambda) = M_{NAP} \times exp^{-S_{NAP} \times (\lambda - \lambda_0)}$; $\lambda_0 = 443$ | $S_{NAP}$ | $M_{NAP}$ |
| $a_{phy}(\lambda)$ | $a_{phy}(\lambda) = \sum Chl\ a_i \times a_{phi}^*$; $a_{phi}^*$ is the spectral shape of each phytoplankton group. | | $Chl\ a_i$ |
| $b_{bp}(\lambda)$ | $b_{bp}(\lambda) = B_{bp} \times (\lambda_0/\lambda)^{S_{bp}}$; $\lambda_0 = 443$ | $S_{bp}$ | $B_{bp}$ |
| Note: $a_{phi}^*(\lambda)$ for 10 different groups of phytoplankton used in this study were extracted from (Dierssen et al., 2006) and Dutkiewicz et al., (2015). | | | |

**Table 3.** Error statistics over all wavelengths and sampling stations (N=301×34=10234; 12 and 22
stations on Sep 29 and Oct 29-30, 2017) from semi-analytical IOP inversion algorithm.

| Parameter | Min. error (%) | Max. error (%) | Mean error (%) | $R^2$(Sep) | $R^2$(Oct) |
|---|---|---|---|---|---|
| $R_{rs}\ \lambda \in [400,700]$ | 0.005 | 40.12 | 18.71 | 0.90 | 0.89 |
| $a_{CDOM}(\lambda),\ \lambda \in [400,700]$ | 0.042 | 11.20 | 5.86 | 0.92 | 0.94 |
| $a_{NAP}(\lambda),\ \lambda \in [400,700]$ | 0.001 | 11.46 | 6.73 | 0.90 | 0.91 |
| $a_{PHY}(\lambda),\ \lambda \in [400,700]$ | 0.001 | 36.42 | 12.19 | 0.84 | 0.85 |
| $b_{bp}(\lambda),\ \lambda = 470$ nm | 0.057 | 40.22 | 10.79 | 0.81 | 0.43 |

**Table 4**. Statistical results between HPLC-measured and NNLS-modeled pigments.

| Pigment | $R^2$ | slope | intercept | RMSE |
|---|---|---|---|---|
| Chl a | 0.963 | 0.878 | 0.099 | 0.125 |
| Chl b | 0.854 | 0.791 | 0.091 | 0.214 |
| Chl $c_1$ | 0.701 | 0.842 | 0.112 | 0.199 |
| Chl $c_2$ | 0.626 | 0.884 | 0.134 | 0.103 |
| Pheophythin a | 0.812 | 0.841 | 0.097 | 0.114 |
| Pheophythin b | 0.783 | 0.632 | 0.112 | 0.145 |
| Peridinin | 0.566 | 0.649 | 0.081 | 0.246 |
| Fucoxanthin | 0.625 | 0.651 | 0.383 | 0.189 |
| Neoxanthin | 0.691 | 0.627 | 0.142 | 0.279 |
| Lutein | 0.742 | 0.651 | 0.109 | 0.298 |
| Violaxanthin | 0.426 | 0.456 | 0.415 | 0.389 |
| Alloxanthin | 0.912 | 0.592 | 0.107 | 0.227 |
| Diadinoxanthin | 0.512 | 0.446 | 0.721 | 0.396 |
| Diatoxanthin | 0.401 | 0.423 | 0.693 | 0.423 |
| Zeaxanthin | 0.689 | 0.516 | 0.802 | 0.584 |
| B-carotenoid | 0.648 | 0.469 | 0.216 | 0.241 |




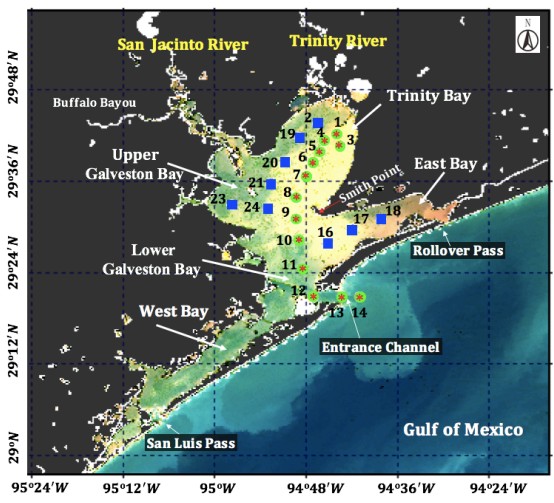


**Figure 1**. Sentinel-3A OLCI RGB image (October 29, 2017) with locations of sampling sites in Galveston Bay acquired on September 29 (red asterisk), October 29 (green circles) and October 30 (blue solid squares), 2017, respectively.


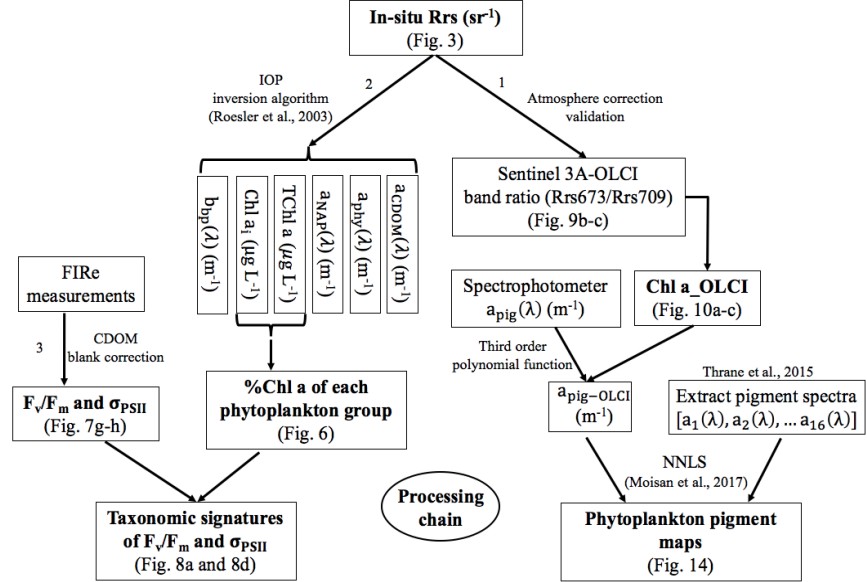

**Figure 2.** Flowchart showing the three processing steps for: (1) retrieving pigments spatial distribution maps from OLCI, (2) distinguishing phytoplankton groups, and (3) assessing phytoplankton physiological parameters and their linkages to taxonomic groups.

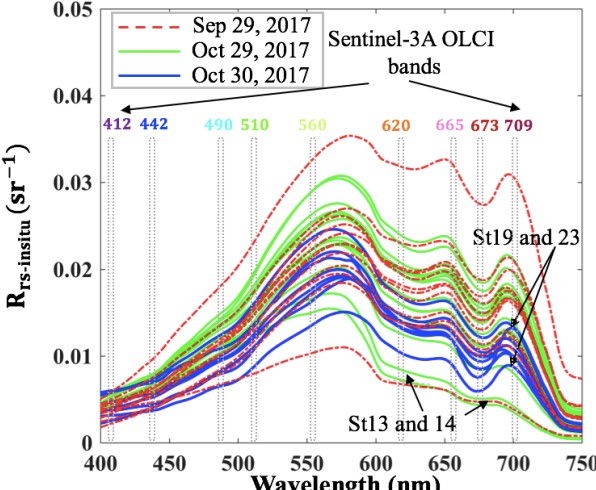

**Figure 3.** $R_{rs\_insitu}$ spectra at stations in GB on September 29, and October 29-30, 2017; vertical bars represent Sentinel-3A OLCI spectral bands.

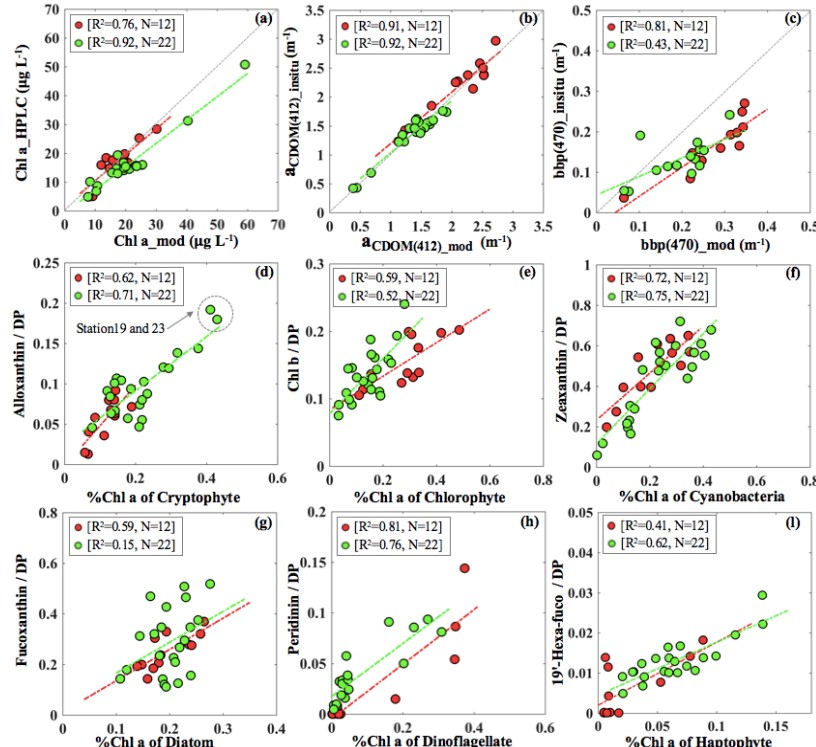

**Figure 4.** (a) Validation of TChl a_mod via HPLC-measured TChl a; individual %Chl a of each detected
taxa versus corresponding %DP shown with (d) cryptophyte, (e) chlorophyte, (f) cyanobacteria, (g)
diatom, (h) dinoflagellate, and (l) haptophyte; red and green dots indicate the samples on September 29
and October 29-30, 2017, respectively. Comparison between in-situ measurements and modeled results
with (b) $a_{CDOM}(412)$ and (c) $b_{bp}(470)$.


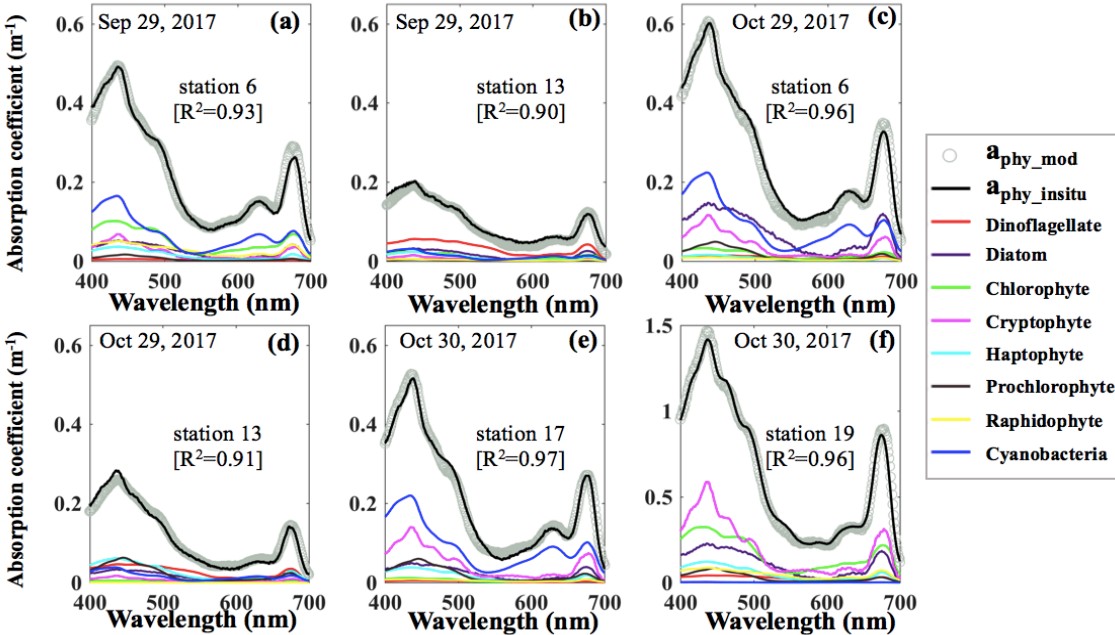

**Figure 5**. Reconstruction of phytoplankton absorption coefficients spectra at station 6 **(a)** and 13 **(b)** on
September 29, 2017, at station 6 **(c)** and 13 **(d)** on October 29, 2017 and at 17**(e)**, and 19 **(f)** on October
30, 2017 based on the mass specific absorption spectra of different phytoplankton groups including
diatom, chlorophyte, dinoflagellate, cryptophyte, cyanobacteria (blue), haptophyte, prochlorophyte and
raphidophyte presented using different colors.

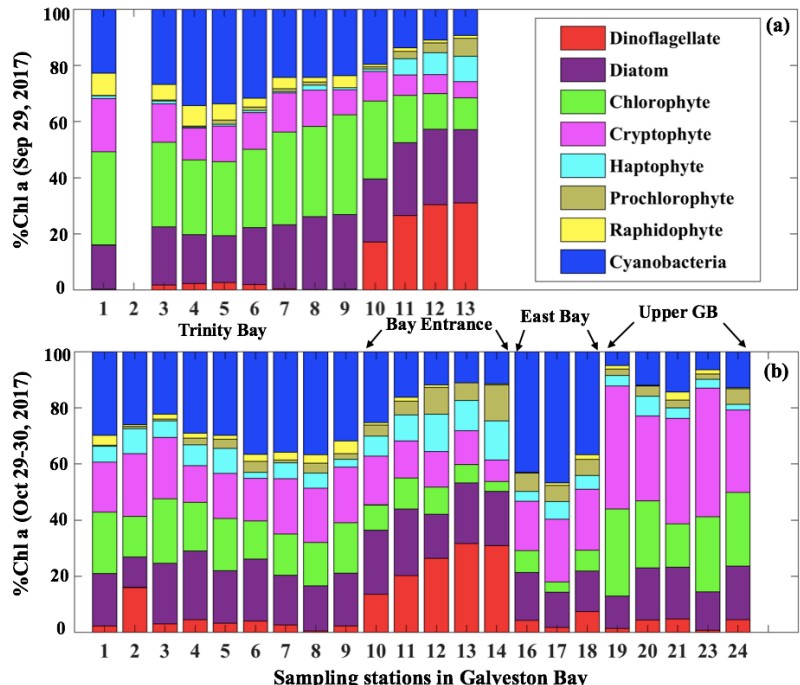

**Figure 6**. Phytoplankton taxonomic compositions detected from IOP inversion algorithm on **(a)**
September 29 and **(b)** October 29-30, 2017 in Galveston Bay; phytoplankton groups are represented in
different colors as shown in the legend.

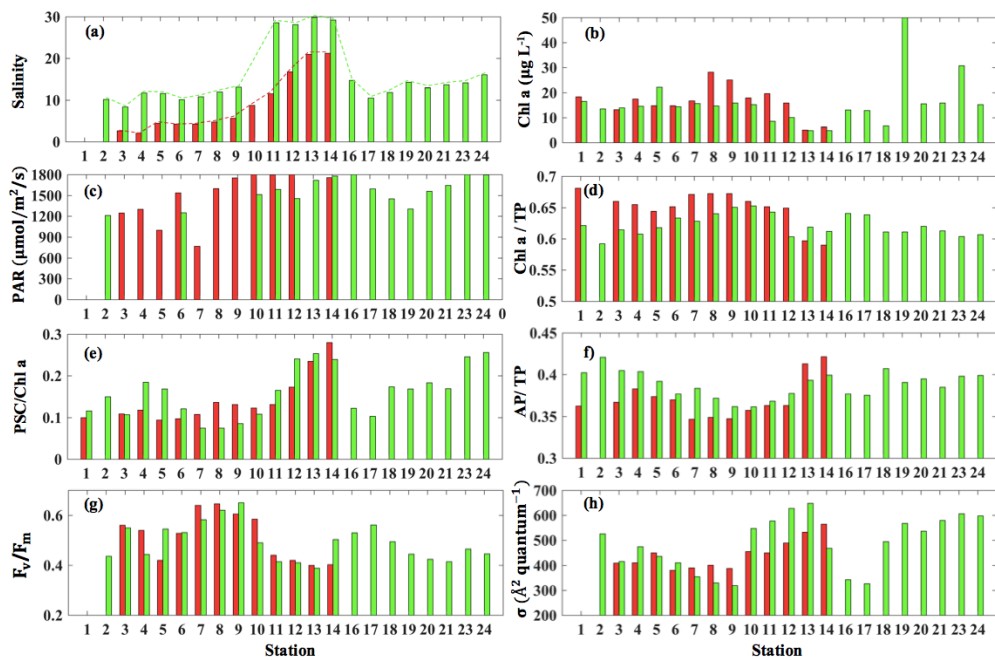


**Figure 7**. Environmental conditions (salinity, light field), pigment composition and physiological state in GB surface waters (red bars indicating samples from September 29, 2017 and blue bars representing samples from October 29 and 30, 2017). **(a)** Salinity, **(b)** Chl-a concentration, **(c)** PAR, **(d)** Chl a/TP, **(e)** PSC/Chl a, **(f)** AP/TP, **(g)** $F_v/F_m$, and **(h)** $\sigma_{PSII}$.

1133
1134

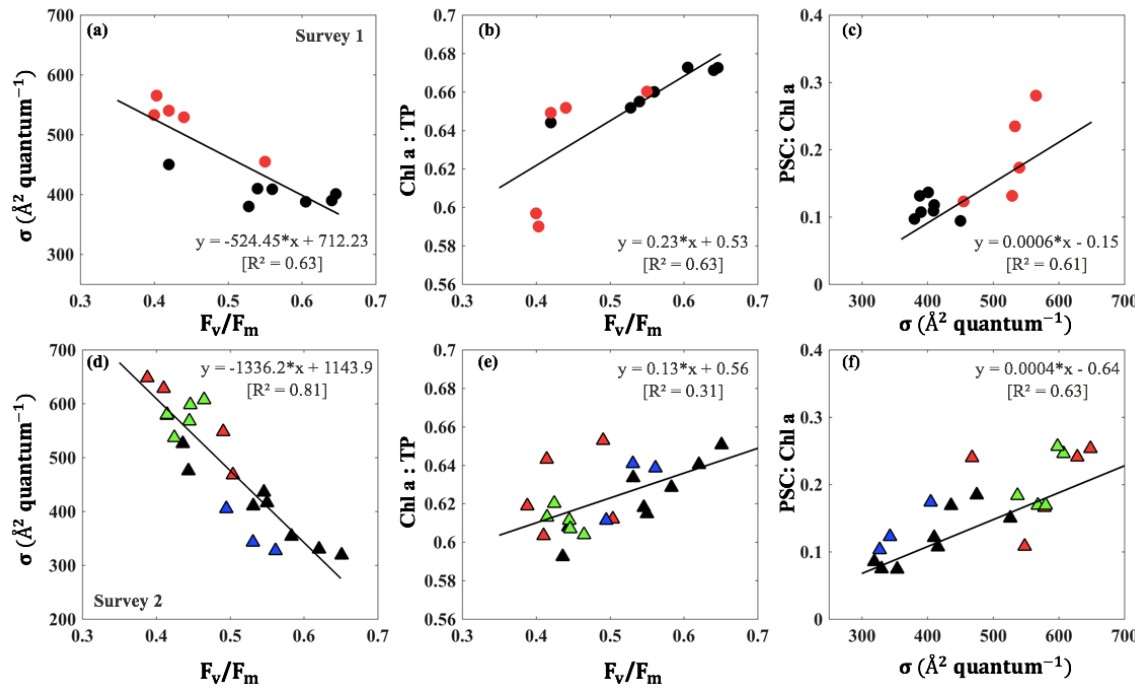

1135

**Figure 8**. **a**, **d)** $\sigma_{PSII}$ against $F_V/F_M$; **b**, **e)** $F_V/F_M$ versus Chl a/TP; and **c**, **f)** $\sigma_{PSII}$ versus PSC/Chl a on September 29 and October 29-30, 2017 respectively. The data points identified by dominant taxa with black, red, green and blue symbols denoting well-mixed, dinoflagellate-haptophyte dominated, cryptophyte-chlorophyte dominated, and cyanobacteria dominated groups, respectively.


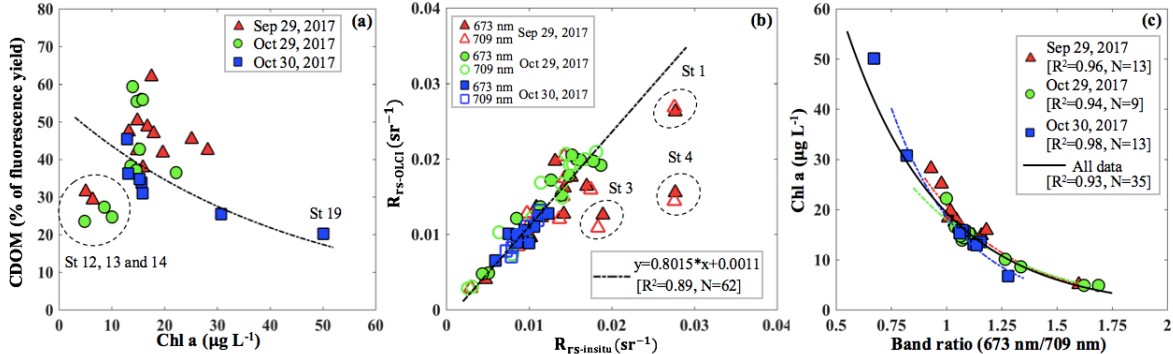


**Figure 9. (a)** Relationship between the percentage of the fluorescence yield of CDOM measured by FIRe
against HPLC measured Chl a concentration. **(b)** Comparisons between $R_{rs\_insitu}$ and $R_{rs\_OLCI}$ at band 9 (673
nm) and band 11 (709 nm). **(c)** Exponential relationships between HPLC-measured Chl a concentrations
and $R_{rs\_insitu}$ band ratio (673 nm/709 nm) in GB on September 29 ($R^2$=0.89), October 29 ($R^2$=0.93) and
October 30 ($R^2$=0.97). Red, green and blue lines and symbols indicate data sets obtained on September 29,
October 29 and 30, 2017 respectively.

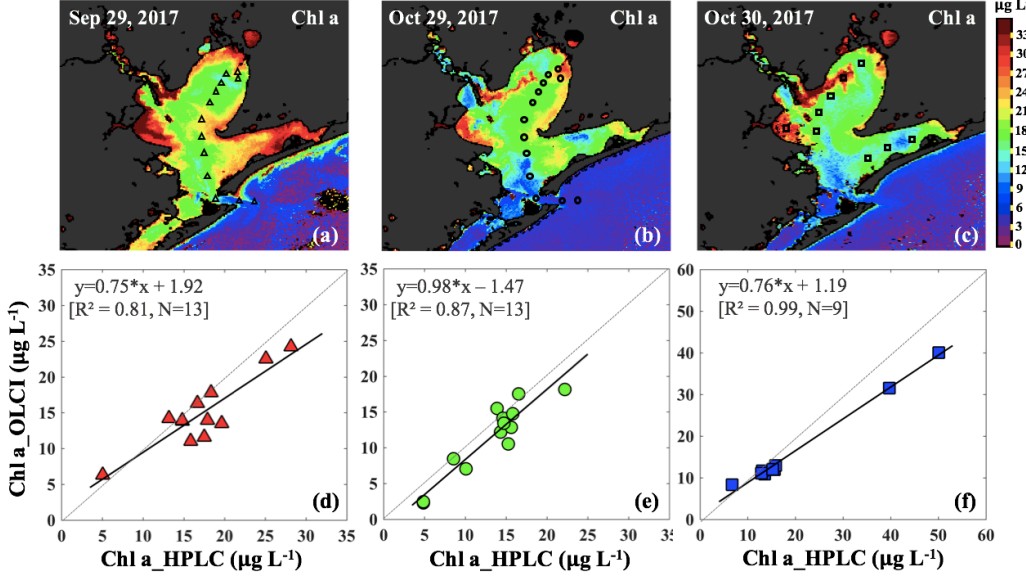


**Figure 10**. Chl a concentration generated based on in-situ band ratio ($R_{rs}673/R_{rs}709$) algorithm with **(a)**,
**(b)** and **(c)** representing Chl a distribution on September 29, October 29 and October 30, 2017,
respectively; **(d)**, **(e)** and **(f)** show the validation between HPLC-measured Chl a and OLCI-derived Chl a
on September 29, October 29 and October 30, 2017, respectively.

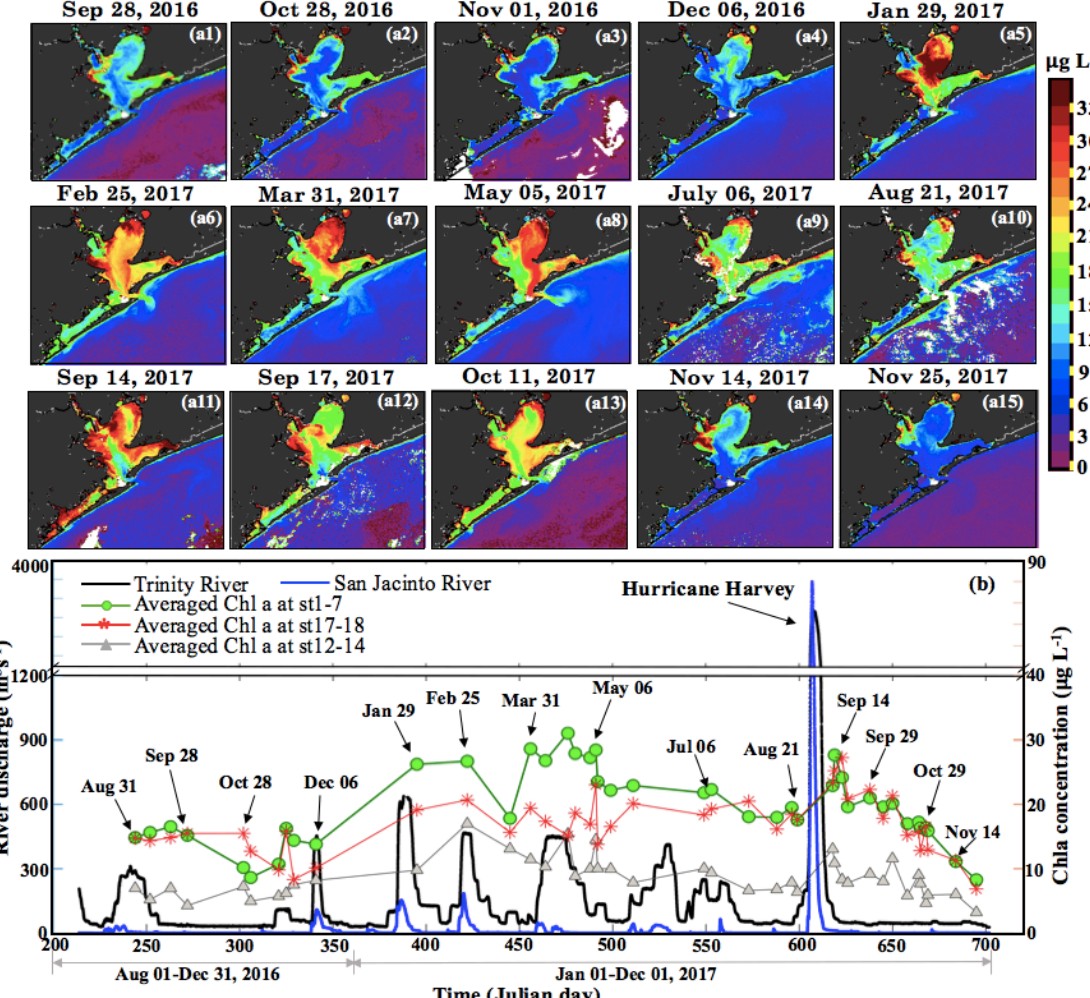


**Figure 11**. (**a₁₋₁₅**) OLCI-derived Chl a shown for the period of August 31, 2016-November 25, 2017. **(b)**
Trinity River discharge at Romayor, Texas (USGS 08066500, black line) and the west flank of the San
Jacinto River (USGS 08067650; blue line); the green, red and gray lines/symbols represent the mean of
Chl a at stations 1-7 in Trinity Bay, at stations 17-18 in East Bay and at stations 12-14 close to the
entrance of GB corresponding to 43 cloud free Sentinel 3A-OLCI images (colored symbols; dated
symbols correspond to images **a₁₋₁₅**).

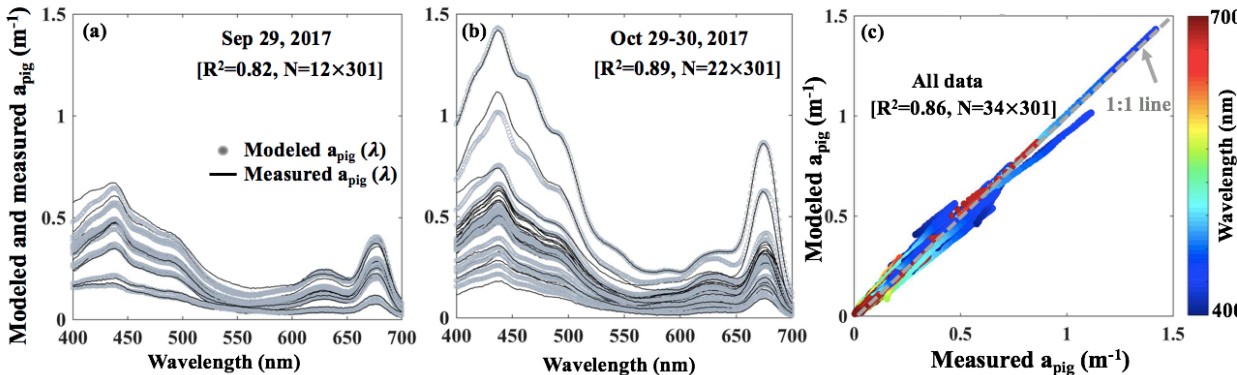


**Figure 12**. Spectrophotometrically measured and multi-regression fitted $a_{pig}(\lambda)$ spectra acquired on **(a)**
September 29 and **(b)** October 29-30, 2017 in GB. Gray and black lines represent modeled and measured
results, respectively. **(c)** Comparison between modeled and spectrophotometrically measured $a_{pig}(\lambda)$ for
all data with color representing wavelength.

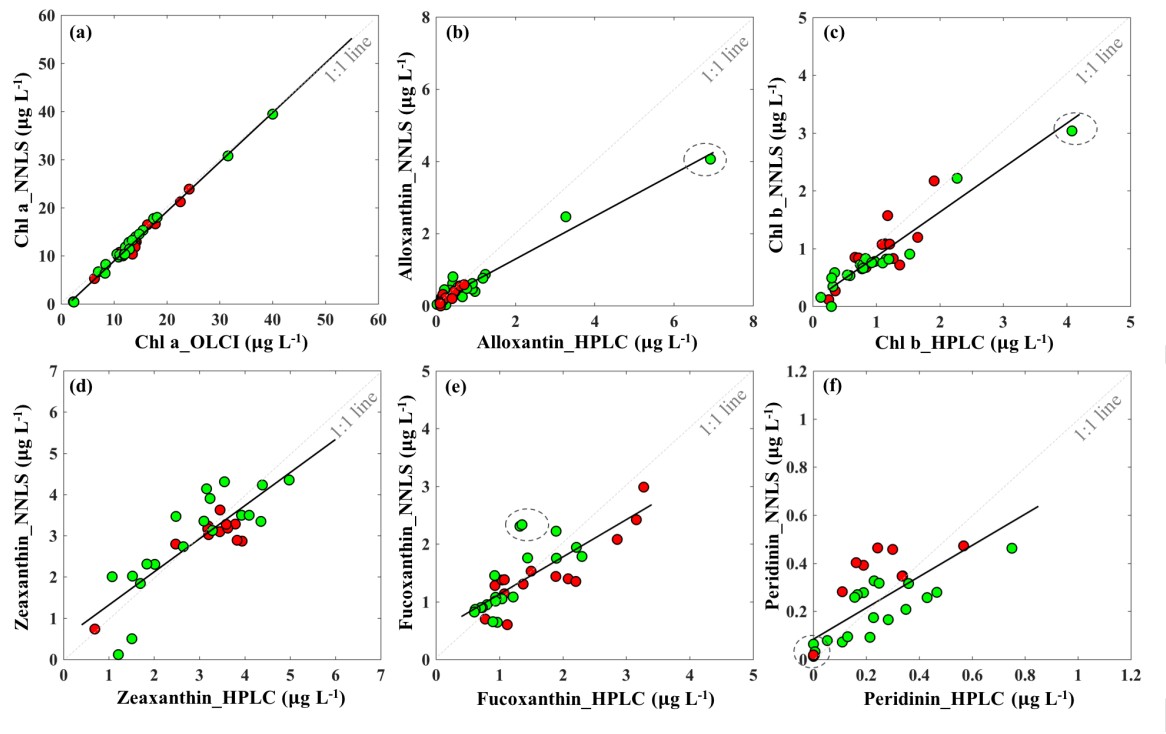


**Figure 13**. Sentinel-3A OLCI derived pigments against HPLC measured pigments in Galveston Bay; **(a)**
Chl a, **(b)** alloxanthin, **(c)** Chl-b, **(d)** zeaxanthin, **(e)** fucoxanthin, and **(f)** peridinin. Red and green
symbols indicate data sets obtained on September 29, and October 29-30, 2017 respectively.

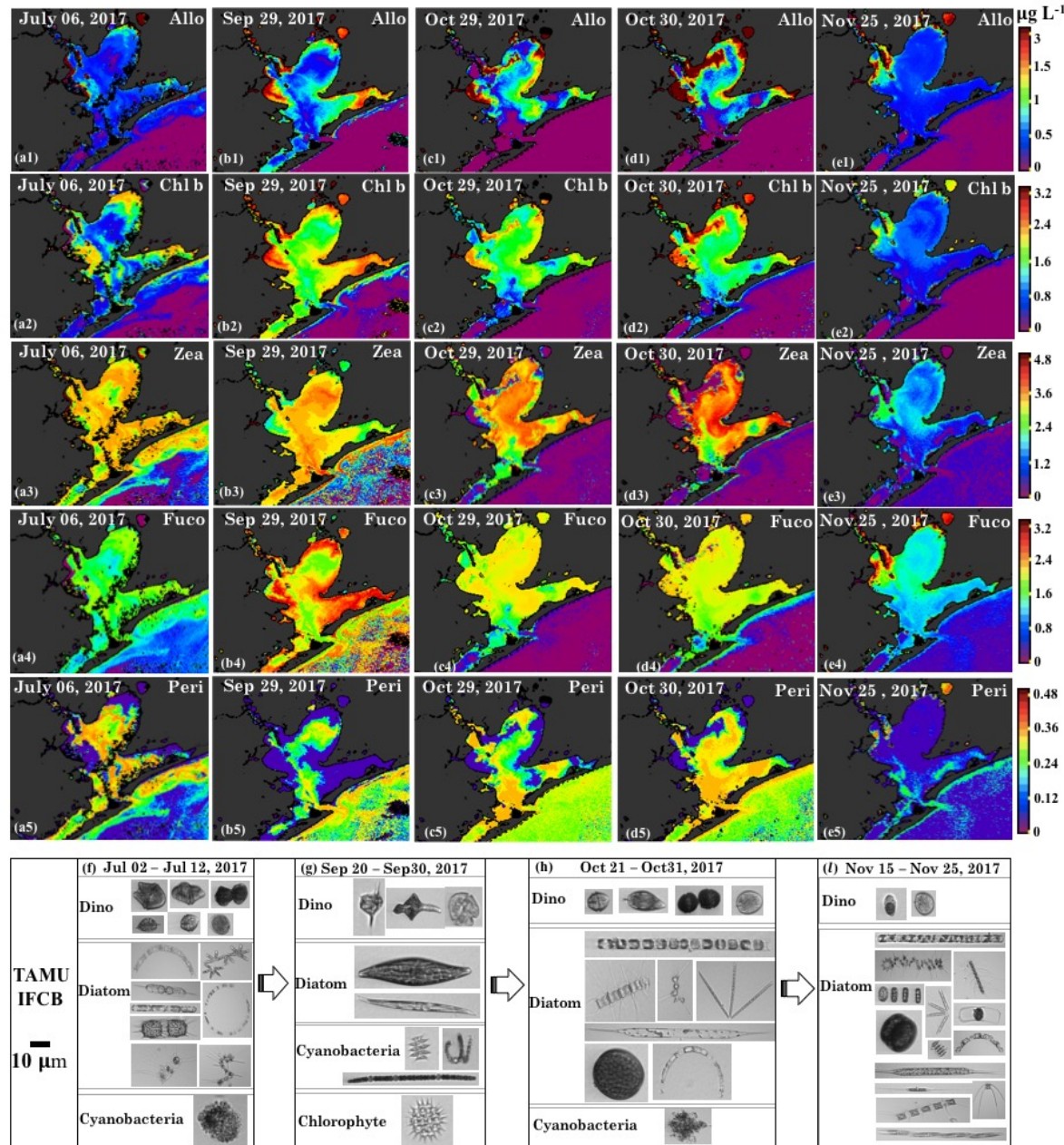

**Figure 14**. Sentinel-3 OLCI derived maps of diagnostic pigments for Galveston Bay. Simulated **a1-e1)**
alloxanthin, **a2-e2)** Chl b, **a3-e3)** zeaxanthin, **a4-e4)** fucoxanthin, and **a5-e5)** peridinin concentrations. a,
b, c, d and e represent columns (maps for July 06, September 29, October 29-30 and November 25, 2017)
and 1-5 represent rows (pigments), respectively; **(f)**, **(g)**, **(h)** and **(l)** are the corresponding IFCB data for
July 06, September 29, October 29-30 and November 25, 2017, respectively; note that IFCB pictures of
fresh water species including chlorophyte and cyanobacteria that appeared on September 20-30, 2017
have been zoomed in for better clarity.