# Peer review of "Floodwater Impact on Galveston Bay Phytoplankton Taxonomy, Pigment Composition and Photo-Physiological State following Hurricane Harvey from Field and Ocean Color (Sentinel-3A OLCI) Observations"

_Biogeosciences, 2018_

## Referee Comment (RC1) · Anonymous Referee #1 · 29 Jan 2019

This paper presents two things: (1) the evolution of the Galveston Bay system following Hurricane Harvey in 2017 drawing on satellite data to do this and (2) the description of the algorithms and their subsequent validation against in situ data.

Papers like this are always difficult to judge as there is a highly technical aspect of algorithm description, justification and validation on the one hand and then the (more interesting to the general readership) description of the evolution of the GB system on

the other.

On balance, the authors have done a good job of algorithm description, justification and validation.

However, in terms of the evolution of the GB system following the hurricane it would be good to draw on actual datasets of river discharge during the event, as well as any other supporting data that could be obtained. There are some statements within the manuscript, such as nutrient loading, which are not substantiated by any data for example. It would also be useful to put a few paragraphs in describing the phytoplankton succession and reasons for it - it would also be worthwhile looking to contrast this event with the background "mean state" i.e. what are the anomalies from other years in the satellite record (this may only be possible for 10 - 20 years for a small subset of variables, such as chlorophyll).

Overall though, a well written manuscript and worthy of publication.

---

## Referee Comment (RC2) · Anonymous Referee #2 · 4 Feb 2019

Summary The authors utilized field data collected after the passage of hurricane Harvey along with OLCI imagery to characterize phytoplankton compositional changes. They use existing methods to invert Rrs to IOPs and the estimating of phytoplankton pigments.

Overall Comments The paper is well written and organized. The methods are well documented and easy to follow.

[Figure]

The errors associated with the phytoplankton pigment retrievals should be more clearly reported, similarly to those of the IOPs.

The paper could be improved by providing greater context surrounding the hurricane. It would be helpful to characterize what the phytoplankton pigments were before the hurricane and for some restoration period after the hurricane (how long did it take for conditions to return to more normal levels?)

Specific Comments Line 151: replace "repetitively" with "repeatedly" Line 281: please reword for clarity and flow

---

## Author Comment (AC1) · 28 Feb 2019

The authors would like to thank Reviewer #1 for the thoughtful and constructive comments that will improve the quality and clarity of the paper. The author's responses are given below in italic font.

**Response to Anonymous Reviewer #1**

"This paper presents two things: (1) the evolution of the Galveston Bay system following Hurricane Harvey in 2017 drawing on satellite data to do this and (2) the description of the algorithms and their subsequent validation against in situ data."

**Response:** *We thank the reviewer for recognizing our efforts in this.*

"Papers like this are always difficult to judge as there is a highly technical aspect of algorithm description, justification and validation on the one hand and then the (more interesting to the general readership) description of the evolution of the GB system on the other. On balance, the authors have done a good job of algorithm description, justification and validation. However, in terms of the evolution of the GB system following the hurricane it would be good to draw on actual datasets of river discharge during the event, as well as any other supporting data that could be obtained. There are some statements within the manuscript, such as nutrient loading, which are not substantiated by any data for example. It would also be useful to put a few paragraphs in describing the phytoplankton succession and reasons for it - it would also be worthwhile looking to contrast this event with the background "mean state" i.e. what are the anomalies from other years in the satellite record (this may only be possible for 10 - 20 years for a small subset of variables, such as chlorophyll)."

**Response:** *Thanks for noting this deficiency. We carefully followed the reviewer's comments to improve the manuscript by providing greater context surrounding the variations before and after hurricane event, which might be more interesting to the general public. We also added more data including river discharge information, OLCI-derived chlorophyll and Imaging FlowCytobot data to support hurricane-induced variations described in this study; these are now shown in Figures 11 and 14. We also appreciate the positive comment on our algorithm description, justification and validation. The corresponding improvements are shown as below:*

(1) *We have included river discharge data from the Trinity and San Jacinto River**s,** for the period August 01, 2016-December 01, 2017 (downloaded from USGS) to understand seasonal patterns of river discharge and the Hurricane Harvey flooding event. Nutrient data for the basin area, is not readily available for GB. We have thus referenced Santschi 1995, who using an extensive set of field observations showed nitrate to be inversely correlated with salinity in GB. Thus, during the flooding event induced by Hurricane Harvey and the extremely high river discharge, nutrients were most likely elevated in GB.*

(2) *A paragraph has been added in **Section 4.3** between line 719-735 in the manuscript to describe the phytoplankton succession and reasons for the variations in phytoplankton abundance and taxonomy.*

(3) *We have not considered using Chl a over the suggested 10-20 year period as the standard MODIS Chl a product is of coarser resolution (1 km) than OLCI (350 m) and likely inaccurate in the optically complex bay waters. Sentinel 3A-OLCI Level 1B data which are only available after June, 2016 were therefore downloaded and processed for Chl a between August 01, 2016-December 01, 2017. Thus, a year-long OLCI-derived Chl a has been used to examine both seasonal and regional variations in Chl*

*a in conjunction with river discharge (Fig. 11). We have also added a section in the revised manuscript: Section 3.2.2 "Long-term Chl a Observations in Comparison with Hurricane Harvey Event".*

[Figure]

(4) **Figure 11**. (**a₁₋₁₅**) OLCI-derived Chl a shown for the period of August 31, 2016-November 25, 2017. (**b**) Trinity River discharge at Romayor, Texas (USGS 08066500, black line) and the west flank of the San Jacinto River (USGS 08067650; blue line); the green, red and gray lines/symbols represent the mean of Chl a at stations 1-7 in Trinity Bay, at stations 17-18 in East Bay and at stations 12-14 close to the entrance of GB corresponding 43 cloud free Sentinel 3A-OLCI images (colored symbols; dated values corresponding to images **a₁₋₁₅**).

(5) *NNLS pigment algorithm has been applied to Chl a maps of July 06, 2017, and November 25, 2017 to further assess the variations of biomarker pigments pre- and post-hurricane. In addition, freely available data of microplankton (10 to 150 μm) recorded by an Imaging FlowCytobot (IFCB) located at the entrance of GB (http://dq-cytobot-pc.tamug.edu/tamugifcb) have been added to Figure 14 to support pigment retrievals.*

[Figure]

**Figure 14**. Sentinel-3 OLCI derived maps of diagnostic pigments for Galveston Bay. Simulated **a₁-e₁)** alloxanthin, **a₂-e₂)** Chl b, **a₃-e₃)** zeaxanthin, **a₄-e₄)** fucoxanthin, and **a₅-e₅)** peridinin concentrations. a, b, c, d and e represent columns (maps for July 06, September 29, October 29-30 and November 25, 2017) and 1-5 represent rows (pigments), respectively; **(f)**, **(g)**, **(h)** and **(l)** are the corresponding IFCB data for July 06, September 29, October 29-30 and November 25, 2017, respectively; note that IFCB pictures of fresh water species including chlorophyte and cyanobacteria that appeared on September 29, 2017 have been zoomed in for better clarity.

"Overall though, a well written manuscript and worthy of publication."

**Response:** *We appreciate that the reviewer found this work interesting and worthy of publication.*

Reference:
Santschi, P. H.: Seasonality in nutrient concentrations in Galveston Bay, Mar. Environ. Res., 40, 337-362, 1995.

---

## Author Comment (AC2) · 28 Feb 2019

The authors would like to thank Reviewer #2 for the valuable comments and suggestions.

**Response to Reviewer #2**

**Overall Comments**

"Summary The authors utilized field data collected after the passage of hurricane Harvey along with OLCI imagery to characterize phytoplankton compositional changes. They use existing methods to invert Rrs to IOPs and the estimating of phytoplankton pigments. The paper is well written and organized. The methods are well documented and easy to follow."

**Response:** *Thanks for the encouraging comments.*

"The errors associated with the phytoplankton pigment retrievals should be more clearly reported, similarly to those of the IOPs."

**Response:** *Thanks for the suggestion. The mean errors of retrieved phytoplankton pigments have been calculated using* $\%error = \left| \frac{X_{modeled} - X_{measured}}{X_{measured}} \right| \times 100$ *and added in Table 4.*

| Pigments | Sep 29, 2017 ($R^2$) | Oct 29, 2017 ($R^2$) | Oct 30, 2017 ($R^2$) | Averaged ($R^2$) | Mean error (%) |
|---|---|---|---|---|---|
| Chl a | 0.95 | 0.97 | 0.98 | 0.97 | 11.36 |
| Chl b | 0.76 | 0.77 | 0.95 | 0.82 | 24.58 |
| Chl $c_1$ | 0.56 | 0.42 | 0.79 | 0.59 | 34.23 |
| Chl $c_2$ | 0.49 | 0.45 | 0.74 | 0.56 | 31.13 |
| Pheophythin a | 0.76 | 0.79 | 0.72 | 0.75 | 17.77 |
| Pheophythin b | 0.75 | 0.88 | 0.76 | 0.79 | 15.65 |
| Peridinin | 0.65 | 0.48 | 0.51 | 0.54 | 42.26 |
| Fucoxanthin | 0.65 | 0.45 | 0.85 | 0.60 | 30.51 |
| Neoxanthin | 0.55 | 0.63 | 0.79 | 0.65 | 31.13 |
| Lutein | 0.61 | 0.78 | 0.72 | 0.70 | 32.54 |
| Violaxanthin | 0.43 | 0.34 | 0.39 | 0.39 | 60.98 |
| Alloxanthin | 0.81 | 0.40 | 0.91 | 0.72 | 32.90 |
| Diadinoxanthin | 0.69 | 0.40 | 0.89 | 0.66 | 48.12 |
| Diatoxanthin | 0.49 | 0.43 | 0.49 | 0.47 | 54.23 |
| Zeaxanthin | 0.76 | 0.65 | 0.78 | 0.73 | 19.03 |
| $\beta$-carotenoid | 0.41 | 0.42 | 0.82 | 0.55 | 44.02 |

"The paper could be improved by providing greater context surrounding the hurricane. It would be helpful to characterize what the phytoplankton pigments were before the hurricane and for some restoration period after the hurricane (how long did it take for conditions to return to more normal levels?)."

(1) **Response:** *Thanks for this very important suggestion. The revised manuscript now includes Chl a evolution in GB before and after the hurricane-induced flooding event along with a new section: **Section 3.2.2** "Long-term Chl a Observations in Comparison with Hurricane Harvey Event". Further,*

*sequence of Sentinel 3A-OLCI derived Chl a and river discharge information for the period of 08/01/2016-12/01/2017 have been made (Fig. 11). The hurricane-induced Chl a variations are clearly observed in Fig. 11 as described in the manuscript between line 530-546.*

[Figure]

**Figure 11**. (**a₁₋₁₅**) OLCI-derived Chl a shown for the period of August 31, 2016-November 25, 2017. (**b**) Trinity River discharge at Romayor, Texas (USGS 08066500, black line) and the west flank of the San Jacinto River (USGS 08067650; blue line); the green, red and gray lines/symbols represent the mean Chl a at stations 1-7 in Trinity Bay, at stations 17-18 in East Bay and at stations 12-14 close to the entrance of GB corresponding to 43 cloud free Sentinel 3A-OLCI images (colored symbols) and dated values corresponding to images **a₁₋₁₅**.

*(2)* **Response:** *The NNLS pigment algorithm has further been applied to two more OLCI-derived Chla maps, one on July 06, 2017 (pre-hurricane) and another one on November, 25, 2017 (normal condition, Fig. 14). In addition, freely available data of microplankton (10 to 150 µm) pictures recorded by an Imaging FlowCytobot (IFCB) at the entrance of GB (http://dq-cytobot-pc.tamug.edu/tamugifcb) have been added to Fig. 14 to support pigment retrievals for July 06, 2017, and November 25, 2017 due to the absence of HPLC measurements on these two days. Pigment maps along with IFCB data both indicate marine dinoflagellates, and cyanobacteria to be dominant before the hurricane on July 06, 2017, whereas, marine diatoms showed dominance in November, 2017, when typical conditions*

*were restored. More detailed results and discussions have been added in Section 3.3.1 and Section 4.3 of the manuscript.*

[Figure]

**Figure 14**. Sentinel-3 OLCI derived maps of diagnostic pigments for Galveston Bay. Simulated **a1-e1)** alloxanthin, **a2-e2)** Chl b, **a3-e3**) zeaxanthin, **a4-e4)** fucoxanthin, and **a5-e5)** peridinin concentrations. a, b, c, d and e represent columns (maps for July 06, September 29, October 29-30 and November 25, 2017) and 1-5 represent rows (pigments), respectively; **(f), (g), (h) and (l)** are the corresponding IFCB data for July 06, September 29, October 29-30 and November 25, 2017, respectively; note that IFCB pictures of fresh water species including chlorophyte and cyanobacteria that appeared on September 20-30, 2017 have been zoomed in for better clarity.

**Specific Comments**

"Line 151: replace "repetitively" with "repeatedly.""

**Response:** *Thank you for pointing this out. We have replaced "repetitively" by "repeatedly" in line 151.*

"Line 281: please reword for clarity and flow."

**Response:** *description of pigment spectra has been revised as below:*
*"where $A(\lambda) = [a_1(\lambda), a_2(\lambda), ... a_n(\lambda)]$ represent mass-specific spectra of 16 pigments (Chl a, Chl b, Chl c1, Chl c2, pheophytin-a, pheophytin-b, peridinin, fucoxanthin, neoxanthin, lutein, violaxanthin, alloxanthin, diadinoxanthin, diatoxanthin, zeaxanthin, and $\beta$-carotenoid), which are the in-vitro pigment absorption spectra over pigment concentrations and can be extracted from supplementary R scripts of Thrane et al. (2015)."*

---

## Author Response (AR1)

**Associate Editor Decision: Publish subject to minor revisions (review by editor)** (17 Mar 2019) by
Maria Tzortziou
Comments to the Author:
Dear Authors,

Thank you for submitting your manuscript for publication in the Biogeosciences Journal.

Your manuscript has been reviewed by two referees. Both referees agreed that the manuscript is very well written and presents original work with strong relevance to the scope of Biogeosciences. Using a combination of field observations and satellite ocean color retrievals, this study provides novel insights on the impacts of extreme events on phytoplankton structure community dynamics in estuarine waters.

The two referees made several suggestions for improvement of the manuscript. These comments have been addressed in the Authors' responses to the referees' comments. As suggested by the reviewers, the errors associated with the phytoplankton pigment retrievals should be more clearly reported in the manuscript. It would be helpful if, in addition to Table 4, you could include some more discussion in paragraph 3.2.4 on the errors in the phytoplankton pigment retrievals, why retrievals for specific pigments (e.g., peridinin) have larger errors, and how they compare with other studies.

Thank you for submitting your manuscript for publication in the Biogeosciences Journal and I look forward to receiving your revised manuscript.

Maria Tzortziou
Editor, Biogeosciences

**Response**: *We wish to thank the editor for her considerations and the constructive comments. Minor revisions have been made. Previously, separate $R^2$ and %error were obtained on September 29, October 29, and 30, 2017, respectively, and further averaged to obtain mean values of $R^2$ and % error. In order to compared the accuracy of satellite-retrieved pigments with other studies as mentioned in the comments, we changed %error to RMSE; in addition, the $R^2$ and RMSE were calculated based on all data acquired during two surveys instead of calculating separately for each day. Further, slope and intercept obtained for different pigments also have been added to Table 4. More discussion has been added to the manuscript in Line 680-715, shown as below. The reason that we added these in section 4.2 instead of section 3.2.4 is that it appears to fit better in the discussion section 4.*

[revised manuscript text omitted]